# A single-chain and fast-responding light-inducible Cre recombinase as a novel optogenetic switch

Hélène Duplus-Bottin[1], Martin Spichty[1†], Gérard Triqueneaux[1], Christophe Place[2], Philippe Emmanuel Mangeot[3], Théophile Ohlmann[3], Franck Vittoz[2], Gaël Yvert[1*]

[1]Laboratory of Biology and Modeling of the Cell, Universite de Lyon, Ecole Normale Superieure de Lyon, CNRS, UMR5239, Universite Claude Bernard Lyon 1, Lyon, France; [2]Laboratory of Physics, Universite de Lyon, Ecole Normale Superieure de Lyon, CNRS, UMR5672, Universite Claude Bernard Lyon 1, Lyon, France; [3]CIRI-Centre International de Recherche en Infectiologie, Universite Claude Bernard Lyon 1, Universite de Lyon, Inserm, U1111, CNRS, UMR5308, Ecole Normale Superieure de Lyon, Lyon, France

**Abstract** Optogenetics enables genome manipulations with high spatiotemporal resolution, opening exciting possibilities for fundamental and applied biological research. Here, we report the development of LiCre, a novel light-inducible Cre recombinase. LiCre is made of a single flavin-containing protein comprising the AsLOV2 photoreceptor domain of *Avena sativa* fused to a Cre variant carrying destabilizing mutations in its N-terminal and C-terminal domains. LiCre can be activated within minutes of illumination with blue light without the need of additional chemicals. When compared to existing photoactivatable Cre recombinases based on two split units, LiCre displayed faster and stronger activation by light as well as a lower residual activity in the dark. LiCre was efficient both in yeast, where it allowed us to control the production of $\beta$-carotene with light, and human cells. Given its simplicity and performances, LiCre is particularly suited for fundamental and biomedical research, as well as for controlling industrial bioprocesses.

**\*For correspondence:**
Gael.Yvert@ens-lyon.fr

**Present address:** † Laboratoire d'Innovation Moleculaire et Applications, Site de Mulhouse – IRJBD, Mulhouse Cedex, France

## Introduction

The wealth of knowledge currently available on the molecular regulations of living systems – including humans – largely results from our ability to introduce genetic changes in model organisms. Such manipulations have been extremely informative because they can unambiguously demonstrate causal effects of molecules on phenotypes. The vast majority of these manipulations were made by first establishing a mutant individual – or line of individuals – and then studying it. This classic approach has two limitations. First, the mutation is present in all cells of the individual. This complicates the analysis of the contribution of specific cells or cell types to the phenotypic alterations that are observed at the whole-organism level. Second, when a mutation is introduced long before the phenotypic analysis, it is possible that the organism has 'adapted' to it either via compensatory regulations or, in case of mutant lines maintained over multiple generations, by compensatory mutations.

For these reasons, other approaches relying on site-specific recombinases were developed to introduce specific mutations in a restricted number of cells of the organism and at a specific time. For instance, the Cre/LoxP system (*Van Duyne, 2015*; *Rajewsky et al., 1996*) consists of two manipulations: a stable insertion, in all cells, of foreign 34 bp DNA sequences called LoxP and the expression of the Cre recombinase in some cells only, where it modifies the DNA by catalyzing recombination between the LoxP sites. The result is a mosaic animal – or plant, or colony of cells – where chromosomal DNA has been rearranged in some cells only. Cre is usually introduced via a

**eLife digest** In a biologist's toolkit, the Cre protein holds a special place. Naturally found in certain viruses, this enzyme recognises and modifies specific genetic sequences, creating changes that switch on or off whatever gene is close by. Genetically engineering cells or organisms so that they carry Cre and its target sequences allows scientists to control the activation of a given gene, often in a single tissue or organ.

However, this relies on the ability to activate the Cre protein 'on demand' once it is in the cells of interest. One way to do so is to split the enzyme into two pieces, which can then reassemble when exposed to blue light. Yet, this involves the challenging step of introducing both parts separately into a tissue.

Instead, Duplus-Bottin et al. engineered LiCre, a new system where a large section of the Cre protein is fused to a light sensor used by oats to detect their environment. LiCre is off in the dark, but it starts to recognize and modify Cre target sequences when exposed to blue light.

Duplus-Bottin et al. then assessed how LiCre compares to the two-part Cre system in baker's yeast and human kidney cells. This showed that the new protein is less 'incorrectly' active in the dark, and can switch on faster under blue light. The improved approach could give scientists a better tool to study the role of certain genes at precise locations and time points, but also help them to harness genetic sequences for industry or during gene therapy.

transgene that is only expressed in the cells to be mutated. The location and orientation of LoxP sites can be chosen so that recombination generates either a deletion, an inversion, or a translocation. Similar systems were developed based on other recombinases/recognition targets, such as Flp/ FRT (*Lee and Luo, 2001*) or Dre-rox (*Anastassiadis et al., 2009*). To control the timing of recombination, several systems were made inducible. Tight control was obtained using recombinases that are inactive unless a chemical ligand is provided to the cells. For example, the widely used Cre-ER$^T$ chimeric protein can be activated by 4-hydroxy-tamoxifen (*Feil et al., 1996*). Other inducible systems rely on chemical-induced dimerization of two halves of the recombinase. For instance, the FKBP–FRB split Cre system consists of two inactive proteins that can assemble in the presence of rapamycin to form a functional recombinase complex (*Jullien et al., 2003*). Similar systems were reported that rendered dimerization of the split Cre fragments dependent on phytohormones (*Weinberg et al., 2019*). Although powerful, these systems present some caveats: ligands are not always neutral to cells and can therefore perturb the biological process under investigation; since they diffuse in tissues, the control of activation is sometimes not precise enough in space and/or time; and the cost or side effects of chemical inducers can be prohibitive for industrial or biomedical applications.

More recently, several authors modified these dimerizing split recombinases to make them inducible by light instead of chemicals. This presents several advantages because (i) light can be used with extreme spatiotemporal precision and high reproducibility; (ii) when applied at low energy, it is neutral to many cell types; and (iii) it is very cheap and therefore scalable to industrial processes. The dimerization systems that were used come from developments made in optogenetics, where various light, oxygen, or voltage (LOV) protein domains have been used as photosensory modules to control transcription (*de Mena et al., 2018*), protein degradation (*Renicke et al., 2013*), dimerization (*Kennedy et al., 2010*; *Strickland et al., 2012*; *Nihongaki et al., 2014*), or subcellular relocalization (*Niopek et al., 2014*; *Witte et al., 2017*). LOV domains belong to the Per-Arnt-Sim (PAS) superfamily found in many sensors. They respond to light via a flavin cofactor located at their center. In the *A. sativa* phototropin 1 LOV2 (AsLOV2) domain, blue light generates a covalent bond between a carbon atom of a flavin mononucleotide (FMN) cofactor and a cystein side chain of the PAS fold (*Crosson and Moffat, 2001*; *Swartz et al., 2001*), resulting in a conformational change including the unfolding of a large C-terminal $\alpha$-helical region called the J$\alpha$ helix (*Swartz et al., 2002*; *Harper et al., 2003*). Diverse optogenetics tools have been developed by fusing LOV domains to functional proteins in ways that made the J$\alpha$ folding/unfolding critical for activity (*Pudasaini et al., 2015*). Among these tools are several photodimerizers that proved useful to control the activity of recombinases. One study reported blue-light-dependent heterodimerization of a split Cre

recombinase using the CIB1-CRY2 dimerizers from the plant *Arabidopsis thaliana* (*Taslimi et al., 2016*) and others successfully used the nMag/pMag dimerizers derived from Vivid (VVD), a protein of the fungus *Neurospora crassa* (*Kawano et al., 2016*; *Sheets et al., 2020*). A third system was based on dimerizers derived from the chromophore-binding photoreceptor phytochrome B (PhyB) of *A. thaliana* and its interacting factor PIF3. In this case, red light was used for stimulation instead of blue light, but the system required the addition of an expensive chemical, the chromophore phycocyanobilin (*Hochrein et al., 2018*).

An ideal inducible recombinase is one that ensures both low basal activity and high induced activity, that is simple to implement, cheap to use, and fast to induce. All dimerizing split Cre systems have in common that two protein units must be assembled in order to form one functional Cre. Thus, the probability of forming a functional recombination synapse, which normally requires four Cre molecules, is proportional to the product of the two units' cellular concentrations to the power of four. Split systems therefore strongly depend on the efficient expression of their two different coding sequences, as previously reported (*Meador et al., 2019*). An inducible system based on a single protein may avoid this limitation. Its implementation by transgenesis would also be simpler, especially in vertebrates.

We report here the development of light-inducible Cre (LiCre), a novel light-inducible Cre recombinase that is made of a single flavin-containing protein. LiCre can be activated within minutes of illumination with blue light, without the need of additional chemicals, and it shows extremely low-background activity in the absence of stimulation as well as high induced activity. Using the production of carotenoids by yeast as a case example, we show that LiCre and blue light can be combined to control metabolic switches that are relevant to the problem of metabolic burden in bioprocesses. We also report that LiCre can be used efficiently in human cells, making it suitable for biomedical research. Since LiCre offers cheap and precise spatiotemporal control of a genetic switch, it is amenable to numerous biotechnological applications, even at industrial scales.

## Results

### The stabilizing N-ter and C-ter α-helices of the Cre recombinase are critical for its activity

A variety of optogenetic tools have been successfully developed based on specific LOV domain proteins possessing α-helices that change conformation in response to light (*Weitzman and Hahn, 2014*). We reasoned that fusing a LOV domain to a helical domain of Cre that is critical for its function could generate a single protein with light-dependent recombinase activity. We searched for candidate α-helices by inspecting the structure of the four Cre units complexed with two LoxP DNA targets (*Ennifar et al., 2003*; *Guo et al., 1997*; *Figure 1a, b*). Each subunit folds in two domains that bind to DNA as a clamp. It was initially reported that helices αA and αE of the amino-terminal domain, as well as helix αN of the C-terminal domain, participate to inter-units contacts (*Guo et al., 1997*); and this role of helix αN was later confirmed (*Ennifar et al., 2003*). Contacts between αA and αE associate all four amino-terminal domains (*Figure 1a*), and contacts involving αN lock the four carboxy-terminal domains in a cyclic manner (*Figure 1b*). These helices were therefore good candidates for manipulating Cre activity. We focused on αA and αN because their location at protein extremities was convenient to design chimeric fusions.

We tested the functional importance of helices αA and αN by gradually eroding them. We evaluated the corresponding mutants by expressing them in yeast cells, where an active Cre can excise a repressive DNA element flanked by LoxP sites and thereby switch ON the expression of a green fluorescent protein (GFP) (*Figure 1c*). After inducing the expression of Cre mutants with galactose, we counted by flow cytometry the proportion of cells that expressed GFP and used this measure to compare recombinase activities of the different mutants (*Figure 1d*). As a control, we observed that the wild-type Cre protein activated GFP expression in all cells under these conditions. Mutants lacking the last two or the last three carboxy-terminal residues displayed full activity. In contrast, mutants lacking four or more of the C-ter residues were totally inactive. This was consistent with a previous observation that deletion of the last 12 residues completely suppressed activity (*Rongrong et al., 2005*). Our series of mutants showed that helix αN is needed for activity and that its residue E340 (located at position −4 from the protein end) is implicated. The role of this glutamic acid is most

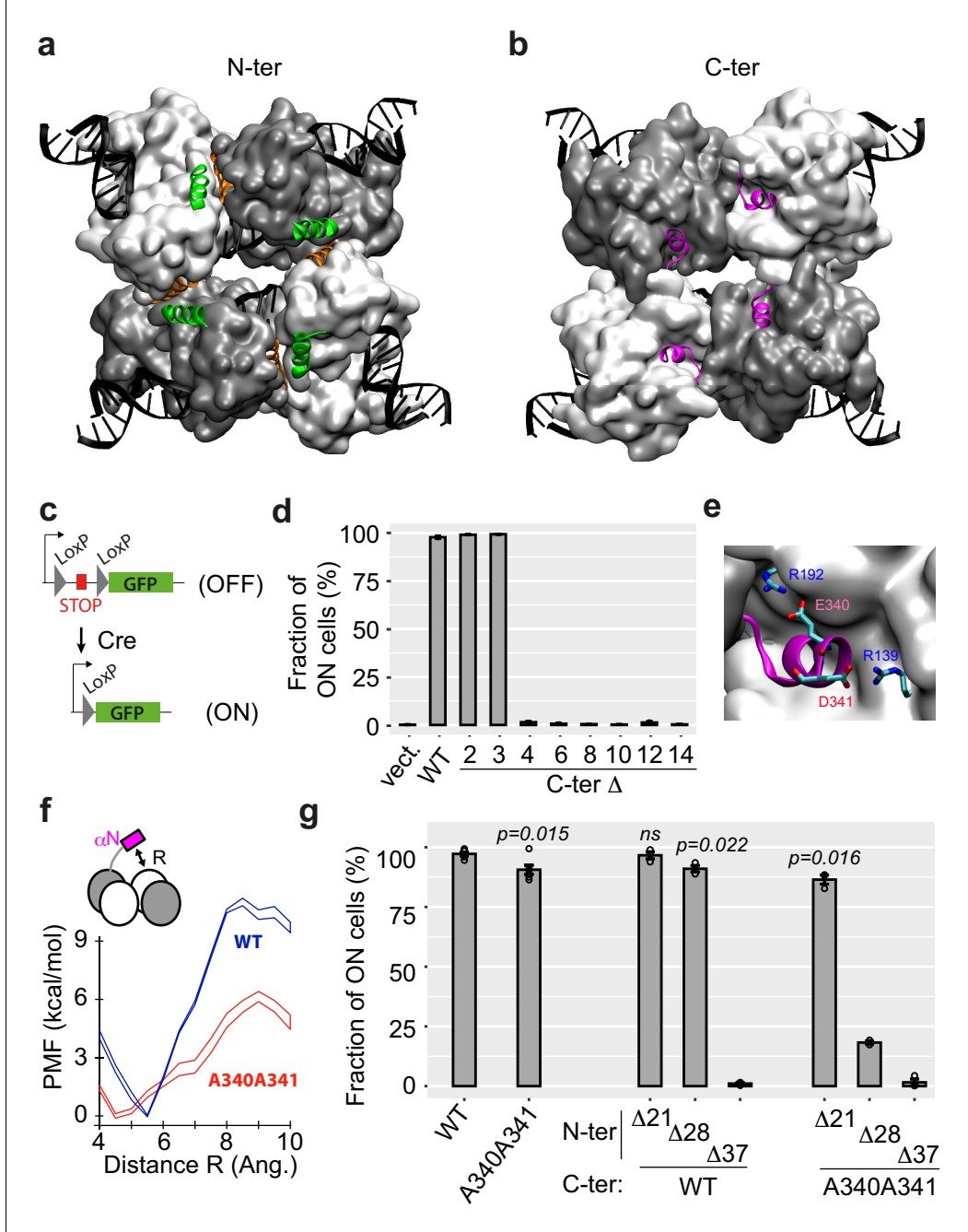

**Figure 1.** N-ter and C-ter α-helices of Cre are critical for activity. (a, b) Structure of the Cre tetramer complexed with DNA (PDB: 1NZB). The four N-ter domains (a) interact via contacts between α-helices A (green) and E (orange), and the four C-ter domains (b) interact via α-helices N (magenta). (c) Yeast reporter system to quantify Cre efficiency. The STOP element includes a selectable marker and a terminator sequence, which prevents expression of the downstream green fluorescent protein (GFP) sequence. (d) Activity of wild-type (WT) and C-ter mutants of Cre measured as the fraction of cells expressing GFP (mean ± s.e.m., *n* = 3 independent transformants). Numbers denote the number of residues deleted from the C-ter extremity. 'Vect': expression plasmid with no insert. (e) Blow-up of αN helix. (f) Energetics of αN displacing (see Materials and methods). PMF: potential mean force (± error defined as $\sigma_{F_i}$ in Materials and methods). (g) Activity of Cre mutants lacking N-terminal residues 2 to X, combined or not with the A340 A341 C-terminal mutation (mean ± s.e.m., *n* = 3 independent transformants). X was 21, 28, or 37 as indicated. P-values: *t*-test of the difference from WT. ns: not significantly different from WT.

The online version of this article includes the following source data for figure 1:

**Source data 1.** Data used to produce *Figure 1d*.

**Source data 2.** Data used to produce *Figure 1f*.
**Source data 3.** Data used to produce *Figure 1g*.

likely to stabilize the complex: the tetramer structure indicates that it interacts with residue R192 of the adjacent unit (*Figure 1e*). Interestingly, although D341 was not essential for activity, the salt bridges of this residue with R139 probably also reinforce attraction between the two protein units. Biomolecular simulations using a simplistic force-field model showed that the free-energy barrier for displacing the αN helix was much lower if E340 and D341 were replaced by alanines (*Figure 1f*). Consistent with this prediction, we observed that a double mutant E340A D341A lost ~10% of activity (*Figure 1g*). This mild (but reproducible) reduction of activity suggested that the double mutation E340A D341A led to a variant of Cre where multimerization was suboptimal.

We also tested the functional importance of α-helix A either in a normal context where the C-terminal part of Cre was intact or where it carried the destabilizing E340A D341A mutation (*Figure 1g*). Deletion of residues 2–37, which entirely ablated helix A, eliminated enzymatic activity (*Figure 1g*). Very interestingly, the effect of shorter deletions depended on the C-terminal context. When the C-terminus was wild-type, removing residues 2–21 (immediately upstream of helix A) had no effect and removing residues 2–28 (partial truncation of αA) decreased the activity by ~10%. When the C-terminus contained the E340A D341A mutation, deletions 2–21 and 2–28 were much more severe, reducing the activity by 12% and 80%, respectively. This revealed genetic interactions between the extremities of the protein, which is fully consistent with a cooperative role of helices αA and αN in stabilizing an active tetramer complex. From these observations, we considered that photo-control of Cre activity might be possible by fusing αA and αN helices to LOV domain photoreceptors.

## Fusions of LOV domains to single-chain Cre confer light-inducible activity

We obtained successful fusions (*Figure 2*) after screening several types of candidate constructs. Our first strategy was to fuse the αN carboxy-terminal helix of Cre to the amino-terminal cap of the LOV domain of protein VVD, a well-characterized photosensor from *N. crassa* (*Nihongaki et al., 2014*; *Zoltowski et al., 2009*; *Vaidya et al., 2011*). The resulting chimeric protein, which contained the full-length Cre connected to VVD via four amino acids, did not display light-dependent recombinase activity (*Figure 2—figure supplement 1*). Introducing the (E340A D341A) double mutation in this chimeric protein reduced the overall activity – as expected from above – but did not cause light dependency. Changing the size of the linker or introducing different mutations at positions 340 and 341 also failed to generate light-dependency (*Figure 2—figure supplement 1*). We noted, however, that the activity of mutants (E340R D341A) and (E340R D341R) was markedly reduced, probably because attraction between E340 and R192 was changed into repulsion between two positively charged arginines.

Our next strategy was based on a modified version of the AsLOV2 domain from *Avena sativa*, which had been optimized and used to build an optogenetic dimerizer by fusing its Jα C-ter helix to the bacterial *SsrA* peptide (*Guntas et al., 2015*). Instead, we fused Jα to the αA amino-terminal helix of Cre. Using the same GFP reporter system as described above for detecting in vivo recombination in yeast, we built a panel of constructs with various fusion positions and directly quantified their activity with and without blue-light illumination.

All fusions displayed reduced activity in both dark and light conditions compared to wild-type Cre. Although variability was high between independent transformants, we identified four constructs – corresponding to fusions of AsLOV2 to residues 17, 19, 27, and 32 of Cre, respectively – where the assay indicated a higher activity after light stimulation (*Figure 2—figure supplement 2*). To confirm this, we recovered the corresponding plasmids from individual yeast transformants, amplified them in bacteria to verify their sequence, and re-transformed them in yeast. This validated a differential activity between dark and light conditions for the constructs corresponding to fusion positions 19, 27, and 32 (*Figure 2a*). Fusion at position 32 (named LOV_Cre32) displayed the highest induction by light, with activity increasing from 15% in dark condition to 50% after 30 min of illumination. Although this induction was significant, a 15% activity of the non-induced form remained too high

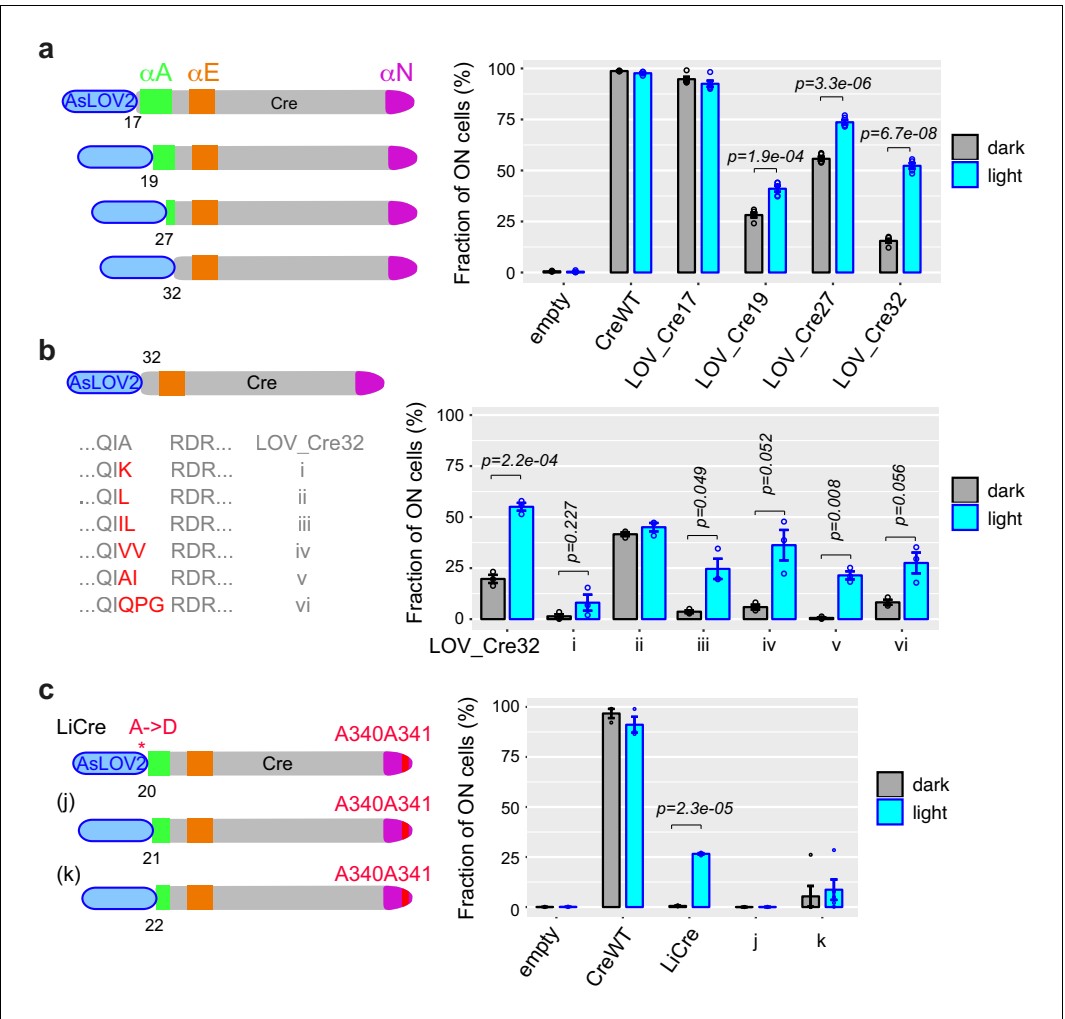

**Figure 2.** Specific single-chain AsLOV2-Cre fusions display photoactivatable recombinase activity. (**a**) Fusions with wild-type Cre. (**b**) Variants of LOV_Cre32 carrying the indicated mutations at the peptide junction. (**c**) Fusions with Cre$^{E340A,D341A}$ mutant. (**a–c**) Numbers indicate the positions on the Cre peptidic sequence where AsLOV2 was fused. All bar plots show recombinase activity measured by flow cytometry (mean ± s.e.m. of the proportion of switched cells, $n$ = 5 independent transformants) after galactose-induced expression of the fusion protein, followed (cyan) or not (gray) by illumination at 460 nm, 36.3 mW/cm$^2$, for 30 min (PAUL LED box). Statistical significance of the differences between dark and light conditions is indicated (*t*-tests). Peptide sequences of all constructs are provided in *Supplementary file 1* – Supplementary Text S2. AsLOV2: *A. sativa* phototropin 1 LOV2; LOV: light, oxygen, or voltage.

The online version of this article includes the following source data and figure supplement(s) for figure 2:

**Source data 1.** Data used to produce *Figure 2a*.
**Source data 2.** Data used to produce *Figure 2b*.
**Source data 3.** Data used to produce *Figure 2c*.
**Source data 4.** Data used to produce *Figure 2—figure supplement 1*.
**Source data 5.** Data used to produce *Figure 2—figure supplement 2*.
**Source data 6.** Data used to produce *Figure 2—figure supplement 3*.
**Figure supplement 1.** Cre-VVD fusions do not display photoactivatable recombinase activity.
**Figure supplement 2.** Systematic analysis of AsLOV2-Cre fusions reveals candidate cases of photoactivatable recombinase activity.
**Figure supplement 3.** Systematic analysis of AsLOV2-Cre$^{A340A341}$ fusions reveals additional cases of photoactivatable recombinase activity.

for most applications. We therefore sought to reduce this residual activity, which we did in two ways.

First, we randomized the residues located at the junction between AsLOV2 and Cre. We used degenerate primers and in vivo recombination (see Materials and methods) to mutagenize LOV_-Cre32 at these positions and directly tested the activity of about 90 random clones. Six of them showed evidence of low residual activity in the dark, and we characterized them further by sequencing and re-transformation. For one clone, photo-induction was not confirmed after re-transformation (*Figure 2b*, clone ii). For the other five clones, residual activity was indeed reduced compared to LOV_Cre32, with the strongest reduction being achieved by an isoleucine insertion at the junction position (*Figure 2b*, clone v). However, this improvement was also accompanied by a weaker induced activity and a larger variability between independent assays.

As a complementary approach to reduce residual activity, we took advantage of the above-described genetic interaction between N-ter truncations and C-ter mutations targeting residues 340 and 341. We built another series of constructs where AsLOV2 fusions to αA helix were combined with the A340A341 double mutation. Results differed from those observed in the context of a wild-type αN (*Figure 2—figure supplement 3*), confirming genetic interaction between N-ter and C-ter mutations. In this assay, two constructs – corresponding to fusion positions 20 and 22 – showed higher recombination activity upon blue-light illumination. We extracted, sequence-verified, and re-tested them together with the one corresponding to fusion position 21. This revealed that fusion at position 20, but not 21 or 22, conferred light-dependent activity (*Figure 2c*). The construct corresponding to fusion of AsLOV2 at position 20 of the Cre A340A341 double mutant displayed a residual activity that was indistinguishable from the negative control and a highly reproducible induced activity of ~25%. Sequencing revealed that it also included a A->D mutation at the fusion junction (*Supplementary file 1* – Supplementary Text S2). We called this construct LiCre and characterized it further.

## Efficiency and dynamics of LiCre photoactivation

We placed LiCre under the expression of the $P_{MET17}$ promoter and tested various lighting conditions. We first used the same illumination system as above, an LED box apparatus commercialized under the name 'PAUL', which was originally designed for viability-PCR assay. This device is suitable for continuous illumination at room temperature. We applied varying intensities and durations on cells that were cultured to stationary phase in the absence of methionine (full LiCre expression). Activity was very low without illumination and increased with both the intensity and duration of light stimulation. The minimal intensity required for stimulation comprised between 0.057 and 1.815 mW/cm$^2$. The highest activity (~65% of switched cells) was obtained with 90 min illumination at 36.3 mW/cm$^2$ (*Figure 3a*). At comparable duration and intensity (30 min and 36.3 mW/cm$^2$), LiCre generated more recombination events in this experiment than in the previous one (*Figure 3a* vs. *Figure 2c*). This improvement likely results from the different expression systems and culture media that were used (induction of the $P_{MET17}$vs. $P_{GAL1}$ promoter). Extending illumination to 180 min did not further increase the fraction of switched cells. Remarkably, we observed that 2 min of illumination was enough to switch 5% of cells, and 5 min illumination generated 10% of switched cells (*Figure 3b*).

We compared these performances with those of two previous systems that were both based on light-dependent complementation of a split Cre enzyme. We constructed plasmids coding for proteins CreN59-nMag and pMag-CreC60 described earlier (*Kawano et al., 2016*) and transformed them in our yeast reporter strain. Similarly, we constructed and tested plasmids coding for the previously described (*Taslimi et al., 2016*) proteins CRY2$^{L348F}$-CreN and CIB1-CreC. All four coding sequences were placed under the control of the yeast $P_{MET17}$ promoter. We analyzed the resulting strains as above after adapting light to match the intensity recommended by the authors (1.815 mW/cm$^2$ for nMag/pMag and 5.45 mW/cm$^2$ for CRY2$^{L348F}$/CIB1). As shown in *Figure 3c*, we validated the photoactivation of nMag/pMag split Cre in yeast, where activity increased about fourfold following 90 min of illumination, but we were not able to observe photoactivation of the CRY2$^{L348F}$/CIB1 split Cre system (*Figure 3d*). In addition, the photoactivation of nMag/pMag split Cre was not as fast as the one of LiCre since 30 min of illumination was needed to observe a significant increase of activity. This observation is consistent with the fact that dimerization of split Cre, which is not required for LiCre, probably limits the rate of formation of an active recombination synapse. Another

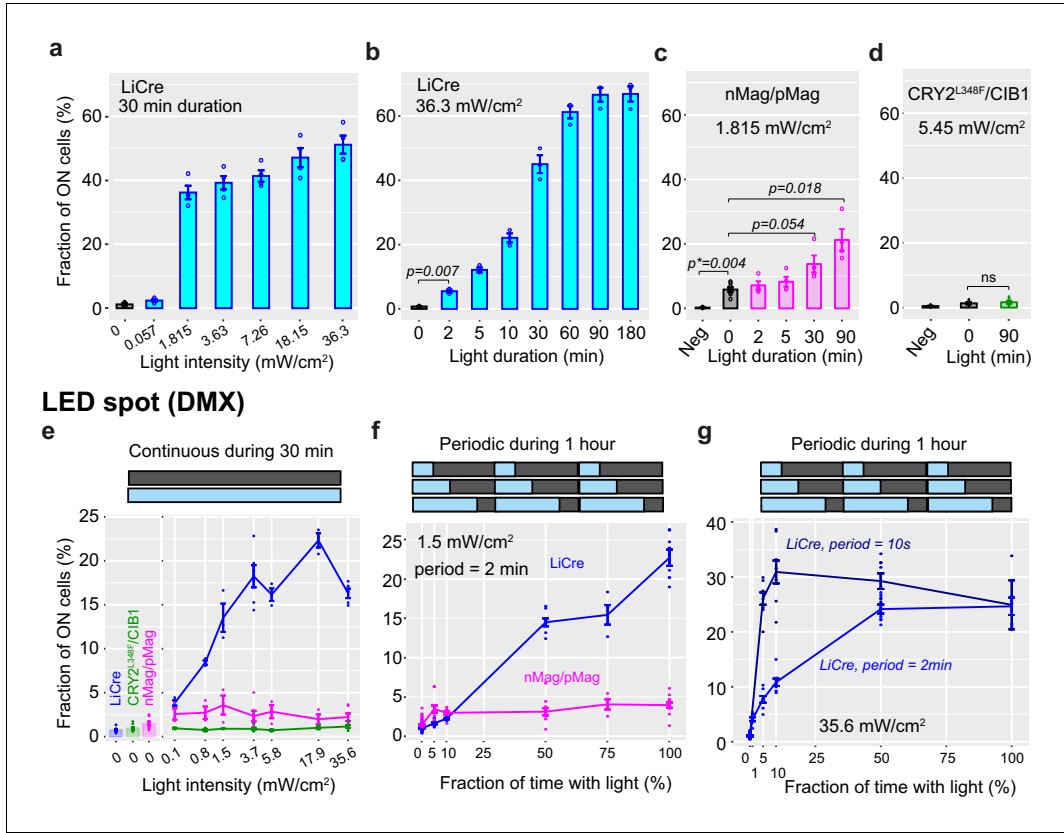

**Figure 3.** Calibration of light induction. (**a, b**) Yeast cells carrying the reporter system of *Figure 1c* and expressing light-inducible Cre (LiCre) were grown to stationary phase and illuminated with blue light (460 nm, PAUL LED box, see Materials and methods) at indicated intensities, then incubated in non-dividing conditions and processed by flow cytometry (mean ± s.e.m., strain GY1761 transformed with pGY466); *n* = 4 and 3 colonies in (**a**) and (**b**), respectively. Illumination conditions varied either in intensity (**a**) or duration (**b**). p: significance from *t*-test (*n* = 3). The fraction of ON cells observed at 0 min was not significantly higher than the fraction of ON GY1761 cells transformed with empty vector pRS314 (p=0.19). (**c**) Yeast strain GY1761 was transformed with plasmids pGY491 and pGY501 to express the two proteins of the nMag/pMag split Cre system (*Kawano et al., 2016*). Cells were processed as in (**b**) with a light intensity that matched the authors' recommendations (1.815 mW/cm$^2$, PAUL LED box). Neg: no illumination, cells containing empty vectors only. p: significance from *t*-tests (*n* = 4); p*: significance from Wilcoxon rank-sum test (*n* = 4). (**d**) Yeast strain GY1761 was transformed with plasmids pGY531 and pGY532 to express the two proteins of the CRY2$^{L348F}$/CIB1 split Cre system (*Taslimi et al., 2016*). Cells were processed as in (**b**) with or without illumination for 90 min at an intensity matching the authors' recommendations (5.45 mW/cm$^2$, PAUL LED box). Neg: no illumination, cells containing empty vectors only. (**e–g**) Yeast strain GY2214 carrying two reporters (green fluorescent protein and mCherry) of Cre-mediated excision was transformed with expression plasmids for LiCre (blue), the nMag/pMag (magenta), or CRY2$^{L348F}$/CIB1 (green) split Cre and illuminated at 30°C under a DMX-controlled LED spot. The fraction of ON cells was computed as the fraction of cells that had activated the mCherry reporter. Illumination was either continuous (**e**) or periodic (**f, g**) at indicated intensity, duration, period, and duty cycle (fraction of period when light is ON).

The online version of this article includes the following source data and figure supplement(s) for figure 3:

**Source data 1.** Data used to produce *Figure 3a*.
**Source data 2.** Data used to produce *Figure 3b*.
**Source data 3.** Data used to produce *Figure 3c*.
**Source data 4.** Data used to produce *Figure 3d*.
**Source data 5.** Data used to produce *Figure 3e*.
**Source data 6.** Data used to produce *Figure 3f*.
**Source data 7.** Data used to produce *Figure 3g*.
**Source data 8.** Data used to produce *Figure 3—figure supplement 1*.
**Figure supplement 1.** Yeast viability after exposure to blue light.

difference was that, unlike LiCre, nMag/pMag split Cre displayed a mild but significant background activity in the absence of illumination (~6% of switched cells) (*Figure 3c*).

We then used another lighting device, a DMX-controlled LED spot, which provided more precise control on both the intensity and dynamics of illumination. We applied varying intensities on reporter cells expressing either LiCre, nMag/pMag split Cre, or CRY2$^{L348F}$/CIB1 split Cre (*Figure 3e*). Absolute efficiencies were weaker than those observed on the PAUL device. This difference is likely due to two factors: the presence of mirrors in the PAUL box (better distribution of light to the cells) and a difference of temperature, which was controlled at 30°C under the DMX spot but not in the PAUL box. Nonetheless, the relative efficiencies were very concordant between the two devices. The DMX spot stimulated an activity of nMag/pMag split Cre that was above background but much weaker than for LiCre, and it did not activate CRY2$^{L348F}$/CIB1 split Cre. This was true at all intensities tested. We observed the highest activation of nMag/pMag split Cre at 1.5 mW/cm$^2$ intensity, which matched the recommendation of its authors (*Kawano et al., 2016*). This calibration also revealed that 0.1 mW/cm$^2$ was enough – although clearly suboptimal – to stimulate LiCre above its residual activity. Intriguingly, we observed a weaker activity at 35.6 mW/cm$^2$ than at 17.9 mW/cm$^2$. However, this difference could be artifactual because it was within the range of variability observed in one of the conditions (3.7 mW/cm$^2$). In parallel, we inspected if blue light at maximal intensity was toxic to yeast cells. We scored colony-forming units (CFUs) from cells exposed or not to 35.6 mW/cm$^2$ blue light (DMX spot) during 1 hr (*Figure 3—figure supplement 1*). We observed no significant differences in the number of CFUs generated by illuminated and non-illuminated samples, nor between LiCre-expressing and non-expressing cells. We conclude that this light exposure is safe regarding yeast viability.

We then interrogated the response of LiCre and nMag/pMag split Cre to periodic stimulations. We expected different response dynamics from these systems because AsLOV2 and VVD have different photocycle lifetimes, with a fast (~80 s) and slow (~18,000 s) decay rate of their adduct (activated) form, respectively (*Pudasaini et al., 2015*). At the optimal energy for nMag/pMag (1.5 mW/cm$^2$), LiCre outperformed nMap/pMag split Cre when light episodes were prolonged but not when they were short (*Figure 3f*). Brief light pulses occurring 2 min apart were insufficient to activate LiCre but fully stimulated nMag/pMag split Cre (*Figure 3f*, 5% of lighting time). This is entirely consistent with a faster decay rate for the AsLOV2 domain of LiCre compared to the VVD domain of nMag/nMag. To specifically and more precisely characterize the response dynamics of LiCre, we applied periodic stimulations using a higher energy (35.6 mW/cm$^2$) and a slow (2 min) or a fast (10 s) period (*Figure 3g*). As expected, the fast regime better stimulated recombination. Remarkably, light pulses of 0.5 s occurring 10 s apart were sufficient to fully stimulate LiCre. Altogether, these results show that, at least in the yeast cellular context, LiCre outperforms the two previous split systems in terms of efficiency, rapidity, and residual background activity.

To demonstrate the control of a biological activity by light, we built a reporter where Cre-mediated excision enabled the expression of the *HIS3* gene necessary for growth in the absence of histidine. We cultured cells carrying this construct and expressing LiCre and spotted them at various densities on two HIS$^-$ selective plates. One plate was illuminated during 90 min while the other one was kept in the dark, and both plates were then incubated for growth. After 3 days, colonies were abundant on the plate that had been illuminated and very rare on the control plate (*Figure 4a*). LiCre can therefore be used to trigger cell growth with light.

We then sought to observe the switch in individual cells. To do so, we replaced GFP by mCherry in our reporter system so that the excitation wavelength of the reporter did not overlap with stimulation of LiCre. We expressed and stimulated LiCre (90 min at 3.63 mW/cm$^2$) in cells carrying this reporter and subsequently imaged them over time. As expected, we observed the progressive apparition of mCherry signal in a fraction of cells (*Figure 4b, c*).

Although convenient for high-throughput quantifications, reporter systems based on the de novo production and maturation of fluorescent proteins require a delay between the time of DNA excision and the time of acquisition. We wished to bypass this limitation and directly quantify DNA recombination. For this, we designed oligonucleotides outside of the region flanked by LoxP sites. The hybridization sites of these primers are too distant for efficient amplification of the non-edited DNA template, but, after Cre-mediated excision of the internal region, these sites become proximal and PCR amplification is efficient (*Figure 4d*). We mixed known amounts of edited and non-edited genomic DNA and performed real-time qPCR to build a standard curve that could be used to infer the

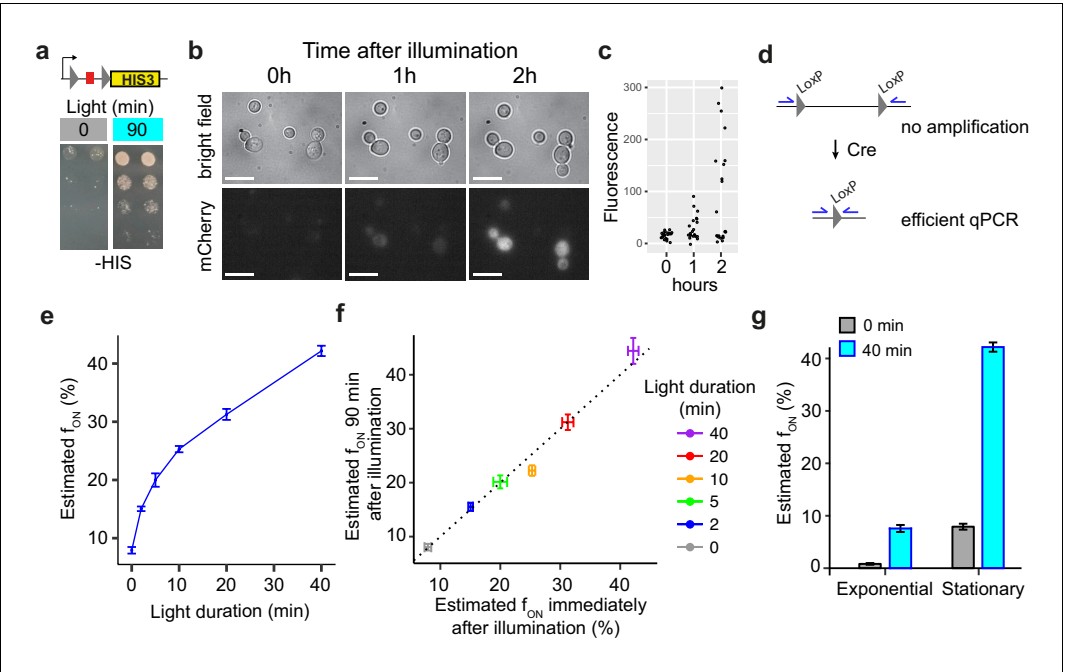

**Figure 4.** Additional functional properties of light-inducible Cre (LiCre). (**a**) Yeast cells expressing LiCre from plasmid pGY466 and carrying an integrated reporter conferring prototrophy to histidine were spotted on two His-plates at decreasing densities. Prior to incubation at 30°C, one plate (right) was illuminated for 90 min at 3.63 mW/cm² intensity (PAUL LED box). (**b**) Time-lapse imaging of yeast cells expressing LiCre and carrying an mCherry reporter of Cre-mediated recombination (strain GY2033 with plasmid pGY466). Cells were grown to stationary phase, illuminated for 90 min at 3.63 mW/cm² intensity, immobilized on bottom-glass wells in dividing condition, and imaged at the indicated time. Bar: 10 μm. (**c**) Quantification of intracellular mCherry fluorescence from (**b**), n = 22 cells. (**d**) Design of qPCR assay allowing to quantify recombination efficiency. (**e**) Quantification of excision by qPCR immediately after illumination at 3.63 mW/cm² intensity (stationary phase, strain GY1761 transformed with pGY466, PAUL LED box). 'Estimated $f_{ON}$' values are the estimated fraction of DNA molecules that underwent recombination, based on a standard curve (see Materials and methods). (**f**) DNA excision does not occur after illumination. X-axis: same data as shown on Y-axis in (**e**). Y-axis: same experiment but after illumination, cells were incubated for 90 min in dark and non-dividing condition prior to harvest and qPCR. (**g**) Quantification of DNA excision by qPCR on exponentially growing or stationary-phase cells (strain GY1761 transformed with pGY466) illuminated at 3.63 mW/cm² intensity (PAUL LED box). Gray: no illumination. Bars in (**e**–**g**): s.e.m. (n = 10 colonies). The online version of this article includes the following source data for figure 4:

**Source data 1.** Data used to produce *Figure 4c*.
**Source data 2.** Data used to produce *Figure 4e–g*.

proportion of edited DNA from qPCR signals. After this calibration, we applied this qPCR assay on genomic DNA extracted from cells collected immediately after different durations of illumination at moderate intensity (3.63 mW/cm², PAUL device). Results were in full agreement with GFP-based quantifications (*Figure 4e*). Excision of the target DNA occurred in a significant fraction of cells after only 2 min of illumination, and we estimated that excision occurred in about 30% and 40% of cells after 20 and 40 min of illumination, respectively. To determine if DNA excision continued to occur after switching off the light, we re-incubated half of the cells for 90 min in the dark prior to harvest and genomic DNA extraction. The estimated frequency of DNA excision was strikingly similar to the one measured immediately after illumination (*Figure 4f*). This was fully consistent with the fast cycling properties of LiCre described above, with a rapid (tens of seconds) reversal of activated LiCre to its inactive state.

The qPCR assay also allowed us to compare the efficiency of light-induced recombination between cell populations in exponential growth or stationary phase. This revealed that LiCre photo-activation was about fourfold more efficient in non-dividing cells (*Figure 4g*). Although the reasons

for this difference remain to be determined, this increase of LiCre photoactivation at stationary phase makes it particularly suitable for bioproduction applications, where metabolic switching is often desired after the growth phase.

## LiCre provides a light switch for carotenoid production

LiCre offers a way to change the activities of cells without adding any chemical to their environment. This potentially makes it an interesting tool to address the limitations caused by metabolic burden, the natural trade-off between the fitness of host cells, and their efficiency at producing exogenous compounds (*Wu et al., 2016*). We therefore tested the possibility to use LiCre to control the production of a commercial compound with light.

Carotenoids are pigments that can be used as vitamin A precursors, antioxidants, or coloring agents, making them valuable for the food, agriculture, and cosmetics industries (*Mata-Gómez et al., 2014*). Commercial carotenoids are generally produced by chemical synthesis or extraction from vegetables, but alternative productions based on microbial fermentations offer remarkable advantages, including the use of low-cost substrates and therefore a high potential for financial gains. Bioproduction of carotenoids can be achieved by introducing recombinant biosynthesis pathways in host microorganisms, which offers the advantage of a well-known physiology of the host and optimizations by genetic engineering. For these reasons, strategies were previously developed to produce carotenoids in the yeast *Saccharomyces cerevisiae*. Expressing three enzymes (*crtE*, *crtI*, and *crtYB*) from *Xanthophyllomyces dendrorhous* enabled *S. cerevisiae* to efficiently convert farnesyl pyrophosphate (FPP) into $\beta$-carotene (*Verwaal et al., 2007*). FPP is naturally produced by *S. cerevisiae* from acetyl-CoA and serves as an intermediate metabolite, particularly for the production of ergosterol that is essential for cellular viability (*Figure 5*). This production is associated with a trade-off: redirecting FPP to $\beta$-carotene limits its availability for ergosterol biosynthesis and therefore impairs growth; and its consumption by the host cell can limit the flux toward the recombinant pathway. Consistently, metabolic burden associated with carotenoids production was shown to be substantial (*Verwaal et al., 2010*). A promising way to deal with this trade-off would be to favor the flux toward ergosterol during biomass expansion and, after enough producer cells are obtained, switch the demand in FPP toward $\beta$-carotene. We therefore explored if LiCre could offer this possibility.

First, we tested if LiCre could allow us to switch ON the exogenous production of carotenoids with light. If so, one could use it to trigger production at the desired time of a bioprocess. We constructed a *S. cerevisiae* strain expressing only two of the three enzymes required for $\beta$-carotene production. Expression of the third enzyme, a bifunctional phytoene synthase and lycopene cyclase, was blocked by the presence of a floxed terminator upstream of the coding sequence of the *crtYB* gene (*Figure 5b*). Excision of this terminator should restore a fully functional biosynthetic pathway. As expected, this strain formed white colonies on agar plates, but it formed orange colonies after transformation with an expression plasmid coding for Cre, indicating that $\beta$-carotene production was triggered (*Figure 5c*). To test the possible triggering by light, we transformed this strain with a plasmid encoding LiCre and selected several transformants, which we cultured and exposed – or not – to blue light before spotting them on agar plates. The illuminated cultures became orange while the non-illuminated ones remained white. Plating a dilution of the illuminated cell suspension yielded a majority of orange colonies, indicating that LiCre triggered *crtYB* expression and $\beta$-carotene production in a high proportion of plated cells (*Figure 5c*). We quantified bioproduction by dosing total carotenoids in cultures that had been illuminated or not. This revealed that 72 hr after the light switch the intracellular concentration of carotenoids had jumped from background levels to nearly 200 µg/g (*Figure 5d*). Thus, LiCre allowed us to switch ON the production of carotenoids by yeast using blue light.

We then tested if LiCre could allow us to switch OFF with light the endogenous ergosterol pathway that competes with carotenoid production for FPP consumption. The first step of this pathway is catalyzed by the Erg9p squalene synthase. Given the importance of FPP availability for the production of various compounds, strategies have been reported to control the activity of this enzyme during bioprocesses, especially in order to reduce it after biomass expansion (*Asadollahi et al., 2008*; *Xie et al., 2015*; *Tippmann et al., 2016*). These strategies were not based on light but derived from transcriptional switches that naturally occur upon addition of inhibitors or when specific nutrients are exhausted from the culture medium. To test if LiCre could offer a way to switch ERG9 activity with

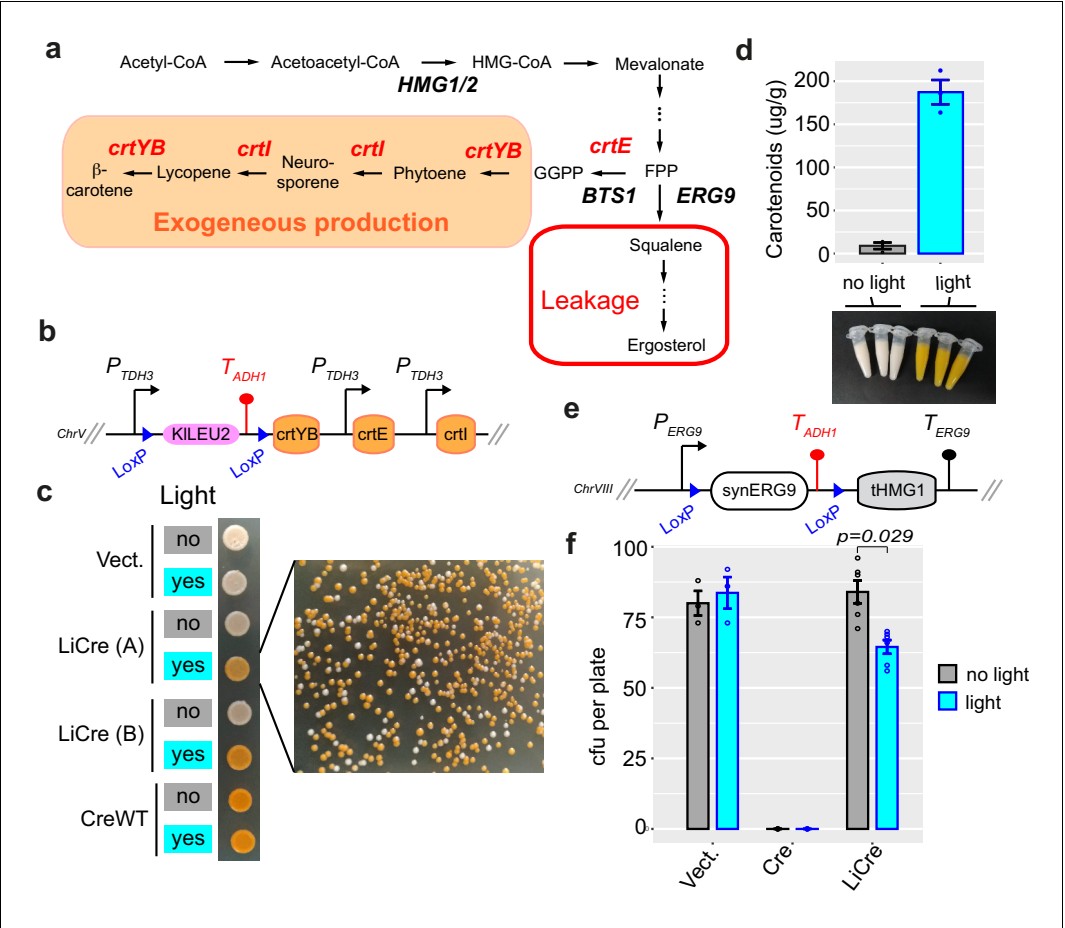

**Figure 5.** Switching ON carotenoid production with light. (**a**) β-Carotene biosynthetic pathway. Exogenous genes from *X. dendrorhous* are shown in red. FPP: farnesyl pyrophosphate; GGPP: geranylgeranyl pyrophosphate. (**b**) Scheme of the switchable locus of yeast GY2247. (**c**) Photoswitchable bioproduction. Strain GY2247 was transformed with either pRS314 (Vect.), pGY466 (light-inducible Cre [LiCre]), or pGY502 (CreWT). Cells were cultured overnight in SD-M-W, and the cultures were illuminated (460 nm, 90 min, 36.3 mW/cm$^2$) or not and then spotted on agar plates. A and B correspond to two independent transformants of the LiCre plasmid. Colonies on the right originate from the illuminated LiCre (A) culture. (**d**) Quantification of carotenoids production. Three colonies of strain GY2247 transformed with LiCre plasmid pGY466 were cultured overnight in SD-W-M. The following day, 10 ml of each culture were illuminated as in (**c**) while another 10 ml was kept in the dark. These cultures were then incubated for 72 hr at 30°C. Cells were pelleted (colors of the cell pellets are shown on picture) and processed for quantification (see Materials and methods). Units are micrograms of total carotenoids per gram of biomass dry weight. Bars: mean ± s.e.m., *n* = 3. (**e**) Scheme of the switchable locus of yeast GY2236. (**f**) Light-induced deletion of squalene synthase gene. Strain GY2236 was transformed with either pRS314 (Vect), pGY502 (Cre), or pGY466 (LiCre). Cells were cultured overnight in 4 ml of SD-M-W. A 100-µl aliquot of each culture was illuminated (as in **c**) while another 100-µl aliquot was kept in the dark. A dilution at ~1 cell/µl was then plated on SD-W. Colonies were counted after 3 days. cfu: colony-forming units (mean ± s.e.m., *n* ≥ 3 plates); p: significance from *t*-test.

The online version of this article includes the following source data for figure 5:

**Source data 1.** Data used to produce *Figure 5d*.
**Source data 2.** Data used to produce *Figure 5e*.

---

light, we modified the *ERG9* chromosomal locus and replaced the coding sequence by a synthetic construct comprising a floxed sequence coding for Erg9p and containing a transcriptional terminator, followed by a sequence coding for the catalytic domain of the 3-hydroxy3-methylglutaryl coenzyme A reductase (tHMG1) (*Figure 5e*). This design prepares *ERG9* for a Cre-mediated switch:

before recombination, Erg9p is normally expressed; after recombination, ERG9 is deleted and the tHMG1 sequence is expressed to foster the mevalonate pathway. Given that ERG9 is essential for yeast viability in the absence of ergosterol supplementation (*Fegueur et al., 1991*), occurrence of the switch can be evaluated by measuring the fraction of viable yeast cells prior to and after the induction of recombination. When doing so, we observed that expression of Cre completely abolished viability, regardless of illumination. In contrast, cultures expressing LiCre were highly susceptible to light: they were fully viable in the absence of illumination and lost ~23% of viable cells after light exposure (*Figure 5f*). Thus, LiCre offers the possibility to abolish the activity of the yeast squalene synthase by exposing cells to light.

### LiCre switch in human cells

Beyond yeast, LiCre may also have a large spectrum of applications on multicellular organisms. Therefore, we tested its efficiency in human cells. For this, we constructed a lentiviral vector derived from the simian immunodeficiency virus (SIV) and encoded a human-optimized version of LiCre with a nuclear localization signal fused to its N-terminus. To quantify the efficiency of this vector, we also constructed a stable reporter cell line where the expression of a membrane-located mCherry fluorescent protein could be switched ON by Cre/Lox recombination. We obtained this line by Flp-mediated insertion of a single copy of the reporter construct into the genome of Flp-In 293 cells (*Figure 6a*, see Materials and methods). Our assay consisted of producing LiCre-encoding lentiviral particle, depositing them on reporter cells for 24 hr, illuminating the infected cultures with blue light, and, 28 hr later, observing cells by fluorescence microscopy. As shown in *Figure 6b*, mCherry expression was not detected in non-infected reporter cells. In cultures that were infected but not illuminated, a few positive cells were observed. In contrast, infected cultures that had been exposed to blue light contained numerous positive cells. We sought to quantify this photoactivation and compare it to the one conferred by nMag/pMag split Cre. For this, we produced two additional sets of lentiviral particles: one encoding the originally described nMag/pMag split Cre (*Kawano et al., 2016*), and one encoding the 'Magnets-opti' coding sequence of 'PA-Cre3.0', where codons had been optimized to avoid intramolecular sequence homology (*Morikawa et al., 2020*). For consistency, all three constructs shared the same vector backbone and CMV promoter. We tested these systems in parallel as above using light energy that was either suited to LiCre (3.6 mW/cm$^2$) or nMag/pMag split Cre (1.8 mW/cm$^2$) and quantified recombination efficiencies by flow cytometry. As shown in *Figure 6c*, the residual activity of Magnets-opti in the dark was very high (~50% of switched cells), which was surprising. LiCre and the original nMag/pMag split Cre system displayed similar levels of induced activity at both low and high light energies; they also showed significant residual activity with 5–10% positive cells for LiCre in the dark and 10–15% for the original nMag/pMag. The highest fold induction was obtained with LiCre at 3.6 mW/cm$^2$, with 5% residual and 31% induced activities, respectively. LiCre can therefore be used to switch genetic activities in human cells with blue light.

## Discussion

By performing a mutational analysis of the Cre recombinase and testing the activity of various chimeric proteins involving Cre variants and LOV domains, we have developed a novel, single-chain, light-inducible Cre recombinase (LiCre). Compared to two previously existing systems relying on light-dependent dimerization of split Cre fragments, LiCre is easier to implement – it requires the expression of a single protein – and it displayed lower background activity in the dark as well as faster and stronger activation by light. LiCre enabled us to use blue light to switch ON the production of carotenoids by yeast and inactivate the yeast squalene synthase. Using a lentiviral vector and human reporter cells, we also showed that LiCre could be used as an optogenetic switch in mammalian systems. We discuss below the possible mechanism of LiCre photoactivation and its properties compared to previously reported photoactivatable recombinases.

### Model of LiCre photoactivation

We built a structural model of LiCre to conceptualize its mode of activation (*Figure 7a*). We based this model on (i) the available structure of the Cre tetramer complexed with its target DNA (*Guo et al., 1997*), (ii) the available structure of AsLOV2 in its dark state (*Guntas et al., 2015*), and

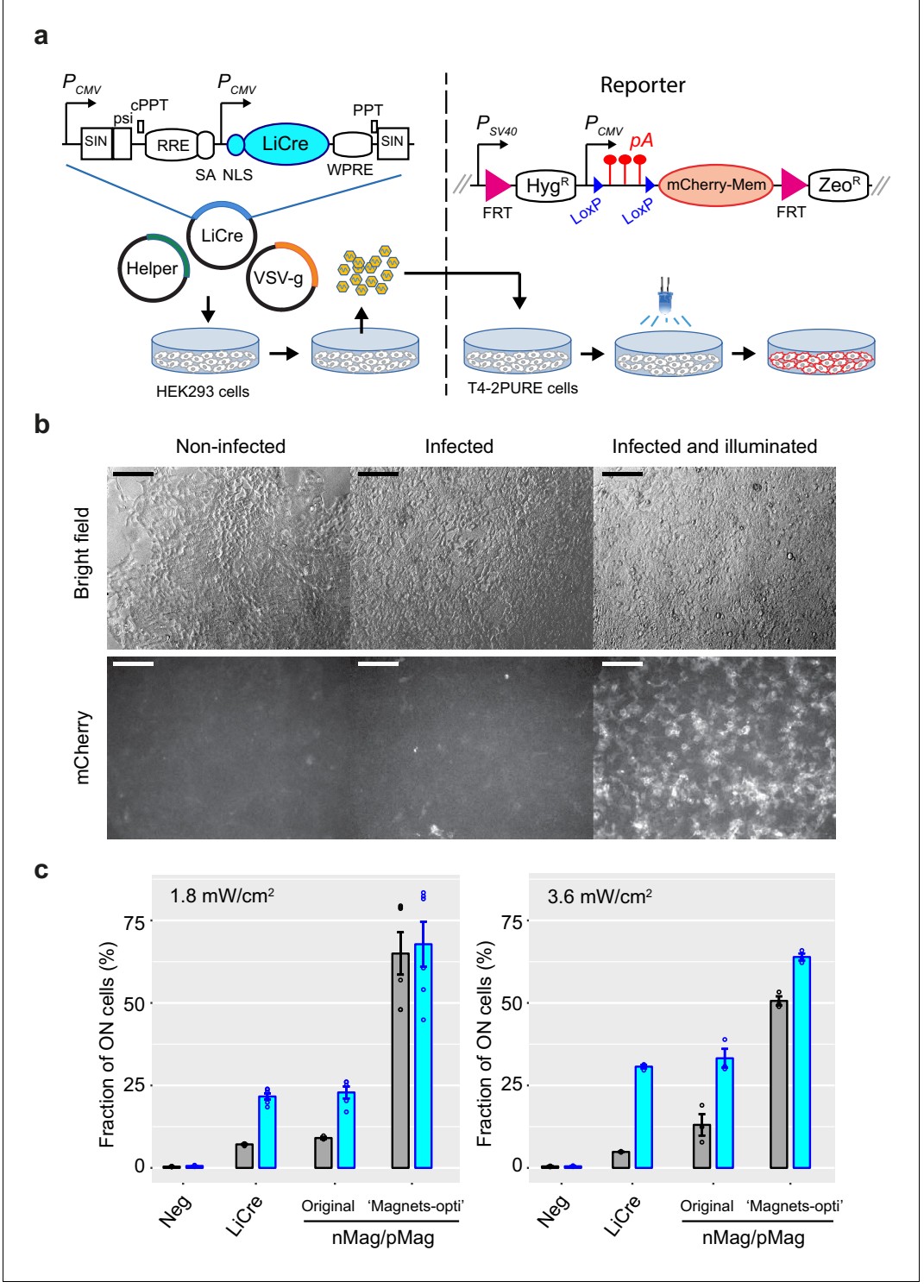

**Figure 6.** Light-inducible Cre (LiCre) photoactivation in human cells. (**a**) Left: lentiviral SIN vector for LiCre expression (plasmid pGY577). $P_{CMV}$: early cytomegalovirus promoter; SIN: LTR regions of simian immunodeficiency virus comprising a partially deleted 3' U3 region, followed by the R and U5 regions; psi: retroviral psi RNA packaging element; cPPT and PPT: central and 3' polypurine tracks, respectively; RRE: Rev/Rev-responsive element; SA: SIV Rev/Tat splice acceptor; NLS: nuclear localization signal; WPRE: woodchuck hepatitis virus regulatory element; Helper: plasmid coding for *gag*, *pol*, *tat*, and *rev*; VSV-g: plasmid encoding the envelope of the vesicular stomatitis virus. Co-transfection in HEK293T cells produces pseudotyped particles. These particles are deposited on T4-2PURE reporter cells, which are then illuminated and imaged. Right: genomic reporter locus of

*Figure 6 continued*

T4-2PURE cells. $P_{SV40}$: promoter from SV40; FRT: FLP recognition targets; Hyg[R]: hygromycin resistance; pA: polyadenylation signal from SV40; Zeo[R]: zeomycin resistance. Recombination between LoxP sites switches ON the expression of mCherry by removing three pA terminators. (b) Microscopy images of T4-2PURE cells following the assay. Bars, 150 µm. All three fluorescent frames were acquired at the same intensity and exposure time. Illumination corresponded to two 20 min exposures at 3.63 mW/cm$^2$ in the PAUL LED box, separated by 20 min without illumination. (c) Flow-cytometry quantification of recombination efficiencies following the assay (mean fraction of mCherry-positive cells ± s.e.m., n = 3 or 6). T4-2 PURE cells were infected with lentiviral particles coding for either LiCre (produced from pGY577), nMag/pMag split Cre (produced from pGY625), or the 'Magnets-opti' version of nMag/pMag split Cre (produced from pGY626). Illumination was conducted as in (b) at the indicated light intensities. Neg: non-infected. We controlled by qPCR that the lower residual activity of LiCre was not due to a lower particle load of the lentiviral preparation (see Materials and methods).

The online version of this article includes the following source data for figure 6:

**Source data 1.** Data used to produce *Figure 6c*, left.
**Source data 2.** Data used to produce *Figure 6c*, right.

(iii) knowledge that the Jα helix of AsLOV2 domains unfolds after light activation (*Swartz et al., 2002*; *Harper et al., 2003*). From this model, we hypothesize that LiCre photoactivation may occur via two synergistic effects. First, the domain AsLOV2 likely prevents Cre tetramerization in the dark state simply because of its steric occupancy. The unfolding of the Jα helix in the light state may allow AsLOV2 to liberate the multimerizing interface. Second, because the Jα helix of AsLOV2 and the αA helix of Cre are immediately adjacent, they may disturb each other in their proper folding so that they cannot exist both simultaneously in their native conformation. The unfolding of Jα in the light state may therefore stimulate proper folding of αA, and thereby allow αA to bind to the adjacent Cre unit. This predicts two possible models of activation of recombinase activity, as depicted in *Figure 7b*. If LiCre is not bound to DNA in its dark state (Model 1), then photoactivation allows it to form a protein dimer bound to each LoxP sites. The two sites can then associate to form the synaptic complex. Alternatively, LiCre may already bind to DNA in the dark state (Model 2). In this case, photoactivation likely reinforces protein–protein interaction of the two bound units and it also allows the two dimers to assemble into a functional recombination synapse. Experiments interrogating LiCre: DNA interactions in dark and light conditions are now needed to test these models.

## LiCre versus other photoactivatable recombinases

Several tools already exist for inducing site-specific recombination with light. They fall in two groups: those that require the addition of a chemical and those that are fully genetically encoded. The first group includes the utilization of photocaged ligands instead of 4-hydroxy-tamoxifen to induce the activity of Cre-ER[T]. This pioneering approach was successful in cultured human cells (*Link et al., 2005*) as well as fish (*Sinha et al., 2010*) and mouse (*Lu et al., 2012*). Later, a more complex strategy was developed that directly rendered the active site of Cre photoactivatable via the incorporation of photocaged amino acids (*Luo et al., 2016*). In this case, cells were provided with non-natural amino acids, such as the photocaged tyrosine ONBY, and were genetically modified in order to express three macromolecules: a specifically evolved pyrrolysyl tRNA synthetase, a pyrrolysine tRNA$_{CUA}$, and a mutant version of Cre where a critical amino acid such as Y324 was replaced by a TAG stop codon. The tRNA synthetase/tRNA$_{CUA}$ pair allowed the incorporation of the synthetic amino acid in place of the nonsense mutation, and the resulting enzyme was inactive unless it was irradiated with violet or ultraviolet light. This strategy successfully controlled recombination in cultured human cells (*Luo et al., 2016*) and zebrafish embryos (*Brown et al., 2018*). We note that it presents several caveats: its combination of chemistry and transgenes is complex to implement, the presence of the tRNA synthetase/tRNA$_{CUA}$ pair can generate off-target artificial C-terminal tails in other proteins by bypassing natural stop codons, and violet/ultraviolet light can be harmful to cells. More recently, a radically different chemical approach was proposed, which consisted of tethering an active TAT-Cre recombinase to hollow gold nanoshells (*Morales et al., 2018*). When delivered to cells in culture, these particles remained trapped in intracellular endosomes. Near-infrared photostimulation triggered activity by releasing the recombinase via nanobubble generation occurring on the particle

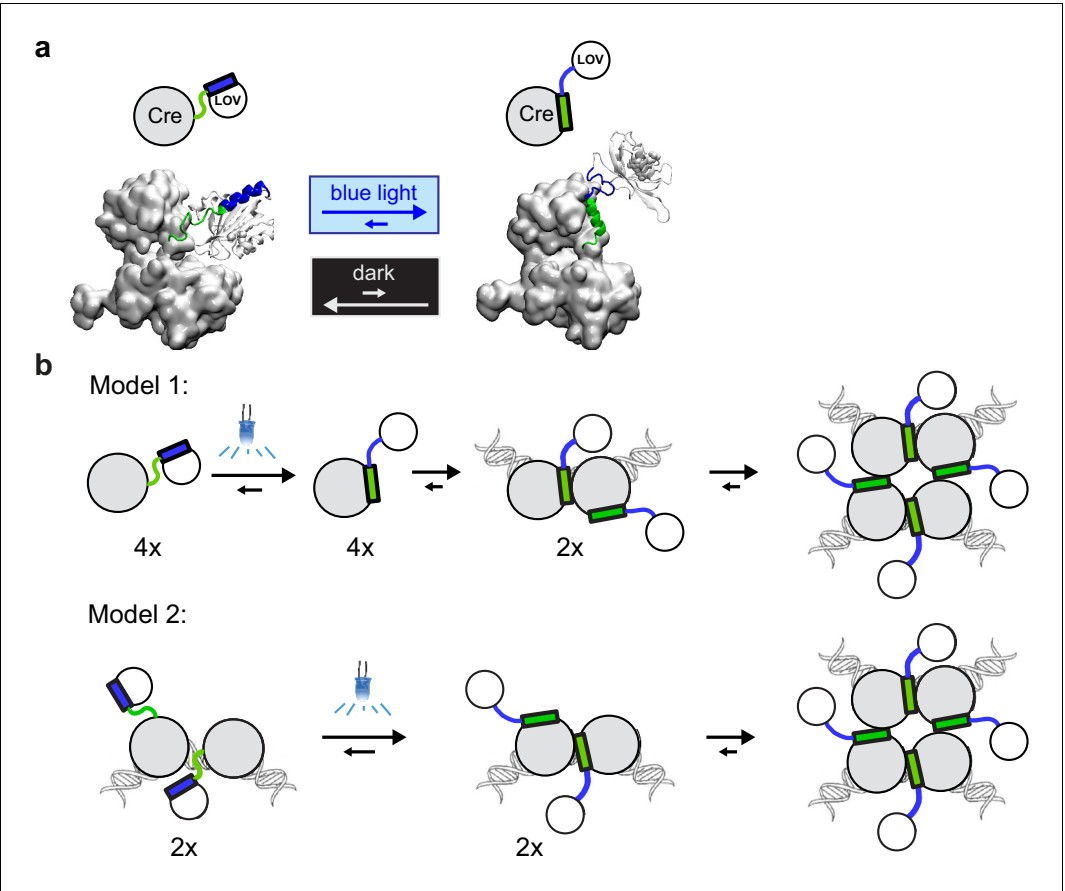

**Figure 7.** Models of light-inducible Cre (LiCre) activation. (**a**) Structural model built using PDB structures 1NZB (Cre) and 4WF0 (AsLOV2). Green: residues of αA helix from Cre. Blue: residues of Jα helix from AsLOV2. (**b**) Proposed models for the photo-induced formation of the LiCre:DNA recombination synapse. Models differ by the affinity of uninduced LiCre for DNA, which may (Model 2) or may not (Model 1) allow their association prior to illumination.

surface. A fourth system is based on the chromophore phycocyanobilin, which binds to the PhyB receptor of *A. thaliana* and makes its interaction with PIF3 dependent on red light. Photostimulation of this interaction was used to assemble split Cre units into a functional complex in yeast (*Hochrein et al., 2018*). A major interest of these last two systems is to offer the possibility to use red light, which is less harmful to cells than blue or violet light and better penetrates tissues. However, all these strategies require to efficiently deliver chemicals to the target cells at the appropriate time before illumination, and their underlying chemistry can be expensive, especially for applications in the context of large volumes such as industrial bioprocesses.

Other systems, such as LiCre, are based on flavin chromophores that are readily available across a wide variety of organisms. This avoids the need of chemical additives. To our knowledge, there are currently three such systems. One is based on the sequestration of Cre between two large photo-cleavable domains (*Zhang et al., 2017*). The principle of light-induced protein cleavage is very interesting, but its application to Cre showed important limitations: a moderate efficiency (~30% of ON cells after the switch), the dependence on a cellular inhibitory chaperone, and the need of violet light. The two other systems are the CRY2/CIB1 and nMag/pMag split Cre (*Taslimi et al., 2016*; *Kawano et al., 2016*) that we evaluated here in comparison to LiCre. An important advantage of LiCre over these systems is that it is made of a single protein. The first benefit of this is simplicity. More efforts are needed to establish transgenic organisms expressing two open reading frames (ORFs) compared to a single one. This is particularly true for vertebrate systems, where inserting several constructs requires additional efforts for characterizing transgene insertion sites and conducting

genetic crosses. For this reason, the two ORFs of the split Cre system were previously combined in a single construct, where they were separated either by an internal ribosomal entry site (IRES) or by a sequence coding a self-cleaving peptide (*Taslimi et al., 2016*; *Kawano et al., 2016*; *Morikawa et al., 2020*). Although helpful, these solutions have important limits: with an IRES, the two ORFs are not expressed at the same level; with a self-cleaving peptide, cleavage of the precursor protein can be incomplete, generating uncleaved products with unknown activity. This was the case for nMag/pMag split Cre in mammalian cells, where a non-cleaved form at ~72 kDa was reported and where targeted modifications of the cleavage sequence increased both the abundance of this non-cleaved form and the non-induced activity of the system (*Morikawa et al., 2020*). The second benefit of LiCre being a single protein is to avoid problems of suboptimal stoichiometry between the two protein units, which was reported as a possible issue for CRY2/CIB1 split Cre (*Meador et al., 2019*). A third benefit is to avoid possible intramolecular recombination between the homologous parts of the two coding sequences. Although not demonstrated, this undesired possibility was suspected for nMag/pMag split Cre because its two dimerizers derive from the same sequence (*Morikawa et al., 2020*). The other advantages of LiCre are its performances. In the present study, we used a yeast-based assay to compare LiCre with split Cre systems. Unexpectedly, although we used the improved version of the CRY2/CIB1 split Cre containing the CRY2-L348F mutation (*Taslimi et al., 2016*), it did not generate photo-inducible recombination in our assay. This is unlikely due to specificities of the budding yeast, such as improper protein expression or maturation, because the original authors reported activity in this organism (*Taslimi et al., 2016*). We do not explain this result, but it is consistent with previous studies reporting that photoactivation of the original version of the CRY2/CIB1 split Cre can be extremely low (*Kawano et al., 2016*), and that the induced activity of the CRY2-L348F/CIB1 system can also be low and highly variable (*Morikawa et al., 2020*). In contrast, we validated the efficiency of nMag/pMag split Cre and so did other independent laboratories (*Morikawa et al., 2020*; *Takao et al., 2020*; *Allen et al., 2019*; *Weinberg et al., 2019*). We observed a significant photoactivation of this system both in yeast and human cells. Recently, the nMag/pMag split Cre system was expressed in mice as a transgene – dubbed PA-Cre3.0 – which comprised the promoter sequence of the chicken beta actin gene (CAG) and the 'Magnets-opti' modified version of the original self-cleaving coding sequence (*Morikawa et al., 2020*). The authors reported that this strategy abolished residual activity and attributed this improvement to a reduction of the expression level of the transgene (*Morikawa et al., 2020*). In our human cell assay, where we used the CMV promoter, the 'Magnets-opti' system instead showed a much higher residual activity than the original nMag/pMag split Cre. This was surprising; it is probably very important to carefully calibrate the expression of Magnets-opti when using it. In contrast, LiCre displayed weaker residual activity than the original nMag/pMag split Cre in the dark. Importantly, LiCre also displayed higher induced activity – especially in yeast – and a faster response to light compared to nMag/pMag split Cre. This strong response probably results from its simplicity since the activation of a single protein involves fewer steps than the activation of two units that must then dimerize to become functional. Finally, we showed that the photocycle of active LiCre is faster than the one of active nMag/pMag split Cre. This offers higher temporal precision for the induction of recombination.

In conclusion, LiCre provides a cheap, simple, low-background, highly efficient and fast-responding way to induce site-specific recombination with light. Given that it works in both yeast and mammalian cells, it opens many perspectives from fundamental and biomedical research to industrial applications.

## Materials and methods

### Strains and plasmids

Plasmids, strains, and oligonucleotides used in this study are listed in *Supplementary file 1* (Supplementary Tables S1, S2, and S3, respectively). LiCre plasmids are available from Addgene (http://www.addgene.org/) under accession numbers 166660, 166661 and 166663.

## Yeast reporter systems

We ordered the synthesis of sequence LoxLEULoxHIS (*Supplementary file 1* – Supplementary Text S1) from GeneCust, who cloned the corresponding *Bam*HI fragment in plasmid pHO-poly-HO to produce plasmid pGY262. The P$_{TEF}$-loxP-KlLEU2-STOP-loxP-spHIS5 construct can be excised from pGY262 by *Not*I digestion for integration at the yeast *HO* locus. This way, we integrated it in a *leu2 his3* strain, which could then switch from LEU+ his- to leu- HIS+ after Cre-mediated recombination (*Figure 4e*). To construct a GFP-based reporter, we ordered the synthesis of sequence LEULoxGreen (*Supplementary file 1* – Supplementary Text S1) from GeneCust, who cloned the corresponding *Nhe*I-*Sac*I fragment into pGY262 to obtain pGY407. We generated strain GY984 by crossing BY4726 with FYC2-6B. We transformed GY984 with the 4 kb *Not*I insert of pGY407 and obtained strain GY1752. To remove the *ade2* marker, we crossed GY1752 with FYC2-6A and obtained strain GY1761. Plasmid pGY537 targeting integration at the *LYS2* locus was obtained by cloning the *Bam*HI-*Eco*RI fragment of pGY407 into the *Bam*HI, *Eco*RI sites of pIS385. Plasmid pGY472 was produced by GeneCust, who synthesized sequence LEULoxmCherry (*Supplementary file 1* – Supplementary Text S1) and cloned the corresponding *Age*I-*Eco*RI insert into the *Age*I, *Eco*RI sites of pGY407. We generated GY983 by crossing BY4725 with FYC2-6A. We obtained GY2033 by transformation of FYC2-6B with a 4 kb *Not*I fragment of pGY472. We obtained GY2207 by transformation of GY983 with the same 4 kb *Not*I fragment of pGY472. To generate GY2206, we linearized pGY537 with NruI digestion, transformed in strain GY855, and selected a LEU+ Lys- colony (pop-in), which we re-streaked on 5-FoA plates for vector excision by counter-selection of URA3 (pop-out) (*Sadowski et al., 2007*). Strain GY2214 was a diploid that we obtained by mating GY2206 with GY2207.

## Yeast expression plasmids

Mutations E340A D341A were introduced by GeneCust by site-directed mutagenesis of pSH63, yielding plasmid pGY372. We generated the N-terΔ21 mutant of Cre by PCR amplification of the P$_{GAL1}$ promoter of pSH63 using primer 1L80 (forward) and mutagenic primer 1L71 (reverse), digestion of pSH63 by *Age*I and co-transformation of this truncated plasmid and amplicon in a *trp1Δ63* yeast strain for homologous recombination and plasmid rescue. We combined the N-terΔ21 and the C-ter E340A D341A mutations similarly, but with pGY372 instead of pSH63. We generated N-terΔ28 and N-terΔ37 mutants, combined or not with C-ter E340A D341A mutations, by the same procedure where we changed 1L71 by mutagenic primers 1L72 and 1L73, respectively.

To generate a Cre-VVD fusion, we designed sequence CreCVII (*Supplementary file 1* – Supplementary Text S1) where the Cre sequence from GENBANK AAG34515.1 was fused to the VVD-M135IM165I sequence from *Zoltowski et al., 2009* via four additional residues (GGSG). We ordered its synthesis from GeneCust and co-transformed it in yeast with pSH63 (previously digested by *Nde*I and *Sal*I) for homologous recombination and plasmid rescue. This generated pGY286. We then noticed an unfortunate error in AAG34515.1, which reads a threonine instead of an asparagine at position 327. We cured this mutation from pGY286 by site-directed mutagenesis using primers 1J47 and 1J48, which generated pGY339 that codes for Cre-4-VVD described in *Figure 2—figure supplement 1*. We constructed mutant C-terΔ14 of Cre by site-directed mutagenesis of pGY286 using primers 1J49 and 1J50, which simultaneously cured the N327T mutation and introduced an early stop codon. Mutants C-terΔ2, C-terΔ4, C-terΔ6, C-terΔ8, C-terΔ10, C-terΔ12 of Cre were constructed by GeneCust, who introduced early stop codons in pGY339 by site-directed mutagenesis. We generated Cre-VVD fusions carrying point mutations at amino acid positions 340 and 341, with or without truncation of the GGSG linker (*Figure 2—figure supplement 1*), by site-directed mutagenesis of pGY339.

To test LOV_Cre fusions, we first designed sequence *Eco*RI-LovCre_chimJa-BstBI (*Supplementary file 1* – Supplementary Text S1) corresponding to the fusion of AsLOV2 with Cre via an artificial α-helix. This helix was partly identical to the Jα helix of AsLOV2 and partly identical to the αA helix of Cre. This sequence was synthesized and cloned in the *Eco*RI and *Bst*BI sites of pSH63 by GeneCust, yielding pGY408. We then generated and directly tested a variety of LOV_Cre fusions. To do so, we digested pGY408 with BsiWI and MfeI and used this fragment as a recipient vector; we amplified the Cre sequence from pSH63 using primer 1G42 as the reverse primer, and one of primers 1M42 to 1M53 as the forward primer (each primer corresponding to a different fusion

position); we co-transformed the resulting amplicon and the recipient vector in strain GY1761, isolated independent transformants, and assayed them with the protocol of photoactivation and flow cytometry described below. We generated and tested a variety of LOV_CreAA fusions by following the same procedure where plasmid pGY372 was used as the PCR template instead of pSH63. A transformant corresponding to LOV_Cre32 and showing light-dependent activity was chosen for plasmid rescue, yielding plasmid pGY415. A transformant corresponding to LOV_CreAA20 was chosen for plasmid rescue, yielding plasmid pGY416. Sanger sequencing revealed that the fusion sequence present in pGY416 was QID instead of QIA at the peptide junction (position 149 on LiCre sequence of *Supplementary file 1* – Supplementary Text S2). All further experiments on LiCre were derived from the fusion protein coded by pGY416.

To introduce random residues at the peptide junction of LOV_Cre32 (*Figure 2b*), we first generated pGY417 using the same procedure as for the generation of pGY415 but with pSH47 instead of pSH63 as the PCR template so that pGY417 has a URA3 marker instead of TRP1. We then ordered primers 1N24, 1N25, and 1N26 containing degenerate sequences, used them with primer 1F14 to amplify the Cre sequence of pSH63, co-transformed in strain GY1761 the resulting amplicons together with a recipient vector made by digesting plasmid pGY417 with NcoI and BsiWI, and isolated and directly tested individual transformants with the protocol of photoactivation and flow cytometry described below. Plasmids from transformants showing evidence of reduced background were rescued from yeast and sequenced, yielding pGY459 to pGY464.

To replace the $P_{GAL1}$ promoter of pGY416 by the $P_{MET17}$ promoter, we digested it with *Sac*I and *Spe*I, PCR-amplified the $P_{MET17}$ promoter of plasmid pGY8 with primers 1N95 and 1N96, and co-transformed the two products in yeast for homologous recombination, yielding plasmid pGY466. We changed the promoter of pGY415 using exactly the same procedure, yielding plasmid pGY465. We changed the promoter of pSH63 similarly using primer 1O83 instead of 1N96, yielding plasmid pGY502.

To express the nMag/pMag split Cre system in yeast, we designed sequence CreN-nMag-NLS-T2A-NLS-pMag-CreC partly (*Supplementary file 1* – Supplementary Text S1) and ordered its synthesis from GeneCust. The corresponding *Bgl*II fragment was co-transformed in yeast for homologous recombination with pGY465 previously digested with *Bam*HI (to remove AsLOV2 and part of Cre), yielding plasmid pGY488 that contained the full system. We then derived two plasmids from pGY488, each one containing one half of the split system under the control of the Met17 promoter. We obtained the first plasmid (pGY491, carrying the TRP1 selection marker) by digestion of pGY488 with SfoI and *Sac*II and co-transformation of the resulting recipient vector with a PCR product amplified from pGY465 using primers 1O80 and 1O82. We obtained the second plasmid (pGY501, carrying the URA3 selection marker) in two steps. We first removed the pMag-CreC part of pGY488 by digestion with *Nde*I and *Sac*II followed by Klenow fill-in and religation. We then changed the selection marker by digestion with *Pfo*I and *Kpn*I and co-transformation in yeast with a PCR product amplified from pSH47 with primers 1O77 and 1O89.

To express the CRY2$^{L348F}$/CIB1 split Cre system in yeast, we designed sequences CIB1CreCter and CRY2CreNter and ordered their synthesis from GeneCust, obtaining plasmids pGY526 and pGY527, respectively. To obtain pGY531, we extracted the synthetic insert of pGY527 by digestion with *Bgl*II and co-transformed it in yeast with the *Nde*I-*Bam*HI fragment of pGY466 for homologous recombination. To obtain pGY532, we extracted the synthetic insert of pGY526 by digestion with *Bgl*II and co-transformed it in yeast with the *Sac*I-*Bam*HI fragment of pSH47 for homologous recombination.

To build a switchable strain for carotene production, we modified EUROSCARF strain Y41388 by integrating a LoxP-KlLEU2-$T_{ADH1}$-LoxP cassette immediately upstream the CrtYB coding sequence of the chromosomally integrated expression cassette described by *Verwaal et al., 2007*. This insertion was obtained by transforming Y41388 with a 6.6 kb BstBI fragment from plasmid pGY559 and selecting a Leu+ transformant, yielding strain GY2247. To obtain pGY559, we first deleted the crtE and crtI genes from YEplac195-YB_E_I (*Verwaal et al., 2007*) by MluI digestion and religation. We then linearized the resulting plasmid with *Spe*I and co-transformed it for recombination in a *leu2Δ* yeast strain with a PCR amplicon obtained with primers 1P74 and 1P75 and template pGY407. After Leu+ selection, the plasmid was recovered from yeast, amplified in bacteria, and verified by restriction digestion and sequencing.

We used CRISPR/Cas9 to build a switchable strain for squalene synthase. We cloned the synthetic sequence gERG9 (*Supplementary file 1* – Supplementary Text S1) in the *Bam*HI-*Nhe*I sites of the pML104 plasmid (*Laughery et al., 2015*) so that the resulting plasmid (pGY553) coded for a gRNA sequence targeting *ERG9*. This plasmid was transformed in GY2226 together with a repair-template corresponding to a 4.2 kb *Eco*RI fragment of pGY547 that contained LoxP-synERG9-T$_{ADH1}$-LoxP with homologous flanking sequences. The resulting strain was then crossed with Y41388 to obtain GY2236.

## Yeast culture media

We used synthetic (S) media made of 6.7 g/l Difco Yeast Nitrogen Base without amino acids and 2 g/l of a powder that was previously prepared by mixing the following amino acids and nucleotides: 1 g of adenine, 2 g of uracil, 2 g of alanine, 2 g of arginine, 2 g of aspartate, 2 g of asparagine, 2 g of cysteine, 2 g of glutamate, 2 g of glutamine, 2 g of glycine, 2 g of histidine, 2 g of isoleucine, 4 g of leucine, 2 g of lysine, 2 g of methionine, 2 g of phenylalanine, 2 g of proline, 2 g of serine, 2 g of threonine, 2 g of tryptophane, 2 g of tyrosine, and 2 g of valine. For growth in glucose condition, the medium (SD) also contained 20 g/l of D-glucose. For growth in galactose condition (induction of P$_{GAL1}$ promoter), we added 2% final (20 g/l) raffinose and 2% final (20 g/l) galactose (SGalRaff medium). Media were adjusted to pH = 5.8 by addition of NaOH 1 N before autoclaving at 0.5 bar. For auxotrophic selections or P$_{MET17}$ induction, we used media where one or more of the amino acids or nucleotides were omitted when preparing S. For example, SD-W-M was made as SD but without any tryptophane or methionine in the mix powder.

## Photoactivation and flow-cytometry quantification of recombinase activity

Unless mentioned otherwise, quantitative tests were done by flow cytometry using yeast reporter strain GY1761. We used two devices for photoactivation. The first device was a PAUL apparatus from GenIUL: a box with mirrors on its walls and floor, and equipped with 460 nm blue LEDs on its ceiling. Using a NovaII photometer (Ophir Photonics), we measured that a 100% intensity on this apparatus corresponded to an energy of 36.3 mW/cm$^2$. We used Zomei ND filters when we needed to obtain intensities that were not tunable on this device. The second device was a computer-controlled DMX LED spot (450 nm) purchased from Neptune-LED (Grenoble, France), which we placed above a thermostated platform. The yeast reporter strain was transformed with the plasmid of interest, pre-cultured overnight in selective medium corresponding to conditions of transcriptional activation of the plasmid-borne Cre construct (SGalRaff-W for P$_{GAL1}$ plasmids, SD-W-M for P$_{MET17}$ plasmids, SD-W-U-M for split Cre systems) with no particular protection against ambient light. The saturated culture was transferred to two 96-well polystyrene flat-bottom Falcon sterile plates (100 µl per well) and one plate was illuminated at the indicated intensities while the other plate was kept in the dark. After the indicated duration of illumination, cells from the two plates were transferred to a fresh medium allowing expression of GFP but not cell division (SD-W-H or SD-W-U-H, strain GY1761 being auxotroph for histidine), and these cultures were incubated at 30°C for 90 min. Cells were then either analyzed immediately by flow cytometry or blocked in phosphate-buffered saline (PBS + 1 mM sodium azide) and analyzed the following day.

We acquired data for 10,000 events per sample using a FACSCalibur (BD Biosciences) or a MACSQuant VYB (Miltenyi Biotech) cytometer after adjusting the concentration of cells in PBS. We analyzed raw data files in the R statistical environment (http://www.r-project.org) using custom-made scripts based on the flowCore package (*Hahne et al., 2009*) from bioconductor (http://www.bioconductor.org). We gated cells automatically by computing a perimeter of (FSC-H, SSC-H) values that contained 40% of events (using 2D-kernel density distributions). We plotted the gates of all samples belonging to a single experiment (usually corresponding to the same day). This occasionally highlighted samples for which the gates were clearly shifted (outliers), indicating cell populations of unexpected size, and we therefore discarded these rare samples for further analysis. A threshold of fluorescent intensity (GFP or mCherry) was set to distinguish ON and OFF cells (i.e., expressing or not the reporter). To do this, we included in every experiment a negative control made of the reporter strain transformed with an empty vector and chose the 99.9th percentile of the corresponding 4000 fluorescent values (gated cells) as the threshold. No statistical power analysis was done

prior to the experiment, no masking of samples IDs was applied during the experiments, and no data randomization was used.

## Quantification of fluorescence levels from microscopy images

For *Figure 3g*, we segmented individual cells on bright-field images using the ImageJ Lasso plugin. Then, we measured on the fluorescence images the mean gray value of pixels in each segmented area, providing single-cell measures of fluorescence. For each image, the background fluorescence level was quantified from eight random regions outside of cells and with areas similar to single cells. This background level was subtracted from the fluorescence level of each cell.

## Quantification of carotenoids from yeast

We strictly followed the procedure described in *Verwaal et al., 2007*, which consists of mechanical cell lysis using glass beads, addition of pyrogallol, KOH-based saponification, and extraction of carotenoids in hexane. Quantification was estimated by optical absorption at 449 nm using a Biowave spectrophotometer. No statistical power analysis was done prior to the experiment, no masking of samples IDs was applied during the experiments, and no data randomization was used.

## Human reporter cell line

We built a reporter construct for Cre-mediated recombination in human cells based on Addgene's plasmids 55779, containing a membrane-addressed mCherry sequence (*Yost et al., 2007*) (mCherry-Mem), and 51269, containing a zsGreen-based reporter of Cre recombination (*Hermann et al., 2014*). Re-sequencing revealed that 51269 did not contain three terminator sequences but only one between the LoxP sites. We applied a multi-steps procedure to (i) restore three terminators, (ii) replace zsGreen with mCherry-Mem, and (iii) have the final reporter in a vector suitable for targeted single-site insertion. First, we inserted a LoxP site between restriction sites *NheI* and *HindIII* of pCDNA5/FRT (Invitrogen) by annealing oligonucleotides 1O98 and 1O99, digesting and cloning this adaptor with NheI and *HindIII*, which yielded plasmid pGY519. Second, we replaced in two different ways the zsGreen sequence of 51269 by the mCherry-Mem sequence of 55779: either by cloning a *SmaI-NotI* insert from 55779 into *EcoRV-NotI* of 51269, yielding plasmid pGY520, or cloning a *EcoRI-NotI* from 55779 into *EcoRI-NotI* of 51269, yielding plasmid pGY521. Third, we inserted the *HindIII-NotI* cassette of pGY520 into the *HindIII-NotI* sites of pGY519, yielding plasmid pGY523. Fourth, we inserted the *HindIII-NotI* cassette of pGY521 into the *HindIII-NotI* sites of pGY519, yielding plasmid pGY524. Fifth, a *HindIII-BamHI* fragment of pGY523 containing one terminator and a *BglII-EcoRI* fragment of pGY524 containing another terminator were simultaneously cloned as consecutive inserts in the *BglII-EcoRI* sites of 51269. Finally, the resulting plasmid was digested with *HindIII* and *BamHI* to produce a fragment that was cloned into the *HindIII-BglII* sites of pGY524 to produce pGY525.

To establish stable cell lines, Flp-In T-REx 293 cells were purchased from Invitrogen (Thermo-Fisher) and transfected with both the Flp recombinase vector (pOG44, Invitrogen) and pGY525. Selection of clonal cells was first performed in medium containing 300 μg hygromycin (Sigma). After 2 weeks, we identified foci of cell clusters, which we individualized by transferring them to fresh wells. One of these clones was cultured for three additional weeks with high concentrations of hygromycin (up to 400 μg) to remove potentially contaminating negative cells. The resulting cell line was named T4-2PURE.

## Lentivirus construct and production

A synthetic sequence was ordered from GeneCust and cloned in the *HindIII-NotI* sites of pCDNA3.1 (Invitrogen V79020). This insert contained an unrelated additional sequence that we removed by digestion with *BamHI* and *XbaI*, followed by blunt-ending with Klenow fill-in. The resulting plasmid (pGY561) encoded LiCre optimized for mammalian codon usage, in-frame with a N-ter located SV40-NLS signal. This NLS-LiCre sequence was amplified from pGY561 using primers Sauci and Flard (Table S3), and the resulting amplicon was cloned in the *AgeI-HindIII* sites of the GAE0 Self-Inactivating Vector (*Mangeot et al., 2004*), yielding pGY577. To obtain a similar vector expressing nMag/pMag split Cre, we ordered its synthetic sequence from GeneCust, who cloned it into the *AgeI-HindIII* sites of pGY577, yielding pGY625. This plasmid was further modified by replacing the *AgeI-*

*Eco*RV portion with a synthetic fragment corresponding to the 'Magnets-opti' coding sequence of 'PA-Cre3.0" (*Morikawa et al., 2020*), yielding pGY626. Sequences of pGY577, pGY625, and pGY626 are provided in *Supplementary file 1* – Supplementary Text S1. Lentiviral particles were produced in Gesicle Producer 293 T cells (TAKARA ref 632617) transiently transfected by an HIV-1 helper plasmid (45% of total DNA), a plasmid encoding the VSV-g envelope (15% of total DNA), and either pGY577, pGY625, or pGY626 (40% of total DNA) as previously described (*Mangeot et al., 2004*). Particle-containing supernatants were clarified, filtered through a 0.45 µm membrane, and concentrated by ultracentrifugation at 40,000 *g* before resuspension in 1× PBS (100-fold concentration). To quantify the particle load of these preparations, we designed a qPCR assay (primers 1Q74,1Q75) and calibrated it on a standard curve obtained by serial dilutions of plasmid DNA. These measures (as mean ± s.e.m. million particles per µl, *n* = 3 replicate measures) on the preparations used in *Figure 6c* were the following: 16 ± 0.7 for LiCre, 8.6 ± 0.4 for nMag/pMag split Cre, and 5.2 ± 0.04 for Magnets-opti.

## LiCre assay in human cells

About $3 \times 10^5$ cells of cell line T4-2PURE were plated in two 6-well plates. After 24 hr, 100 µl of viral particles suspension were added to each well. After another 24 hr, one plate was illuminated with blue light (460 nm) using the PAUL apparatus installed in a 37˚C incubator while the other plate was kept in the dark. For illumination, we applied a sequence of 20 min ON, 20 min OFF under $CO_2$ atmosphere, 20 min ON, where ON corresponded to 3.63 mW/cm$^2$ illumination (or 1.8 mW/cm$^2$ when indicated in *Figure 6c*). Plates were then returned to the incubator and, after 28 hr, were either imaged on an Axiovert135 inverted fluorescent microscope or treated with trypsin, resuspended in 1× PBS, and fixed with paraformaldehyde (PFA) prior to acquisitions on a MACSQuant flow cytometer. Flow-cytometry data was then processed as described above for yeast cells. Four samples were discarded from further analysis because the culture medium had become yellow and because flow cytometry revealed aberrant cell sizes (lower forward scatter [FSC] and very high side scatter [SSC]). Samples IDs were not masked during the experiments. We did not retype the T4-2PURE reporter cell line or the Gesicle Producer cell line, nor did we control them for mycoplasma contamination prior to the experiments because we used them in all the reporter assays that we applied to various constructs and it is extremely unlikely that a contamination would bias results for some constructs and not others.

## Calculation of potential mean force (PMF)

We calculated the free-energy profile (reported in *Figure 1f*) for the unbinding of the C-terminal α-helix in the tetrameric Cre-recombinase complex (*Ennifar et al., 2003*) (PDB entry 1NZB) as follows. The software we used were the CHARMM-GUI server (*Lee et al., 2016*) to generate initial input files; CHARMM version c39b1 (*Brooks et al., 2009*) to setup the structural models and subsequent umbrella sampling by molecular dynamics; WHAM version 2.0.9 (http://membrane.urmc.rochester.edu/content/wham/) to extract the PMF; and VMD version 1.9.2 (*Humphrey et al., 1996*) to visualize structures. To achieve sufficient sampling by molecular dynamics, we worked with a structurally reduced model system. We focused thereby only on the unbinding of the C-terminal α-helix of subunit A (residues 334:340) from subunit F. Residues that did not have at least one atom within 25 Å from residues 333 to 343 of subunit A were removed including the DNA fragments. Residues with at least one atom within 10 Å were allowed to move freely in the following simulations; the remaining residues were fixed to their positions in the crystal structure. For the calculation of the double mutant A340A341, the corresponding residues were replaced by alanine residues. The systems were simulated with the CHARMM22 force field (GBSW and CMAP parameter file) and the implicit solvation model FACTS (*Haberthür and Caflisch, 2008*) with recommended settings for param22 (i.e., cutoff of 12 Å for non-bonded interactions). Langevin dynamics were carried out with an integration time step of 2 fs and a friction coefficient of 4 ps$^{-1}$ for non-hydrogen atoms. The temperature of the heat bath was set to 310 K. The hydrogen bonds were constrained to their parameter values with SHAKE (*Ryckaert et al., 1977*).

The PMF was calculated for the distance between the center of mass of the α-helix (residues 334:340 of subunit A) and the center of mass of its environment (all residues that have at least one atom within 5 Å of this helix). Umbrella sampling (*Torrie and Valleau, 1977*) was performed with 13

independent molecular dynamics simulations where the system was restrained to different values of the reaction coordinate (equally spaced from 4 to 10 Å) using a harmonic biasing potential with a spring constant of 20 kcal mol$^{-1}$ Å$^{-1}$ (GEO/MMFP module of CHARMM). Note that this module uses a pre-factor of ½ for the harmonic potential (as in the case of the program WHAM).

For each simulation, the value of the reaction coordinate was saved at every time step for 30 ns. After an equilibration phase of 5 ns, we calculated for blocks of 5 ns the PMF and the probability distribution function along the reaction coordinate using the weighted histogram analysis method (*Kumar et al., 1992*). A total of 13 bins were used with lower and upper boundaries at 3.75 and 10.25 Å, respectively, and a convergence tolerance of 0.01 kcal mol$^{-1}$. Finally, we determined for each bin its relative free energy $F_i = -kT \ln(\bar{p}_i)$, where $k$ is the Boltzmann constant, $T$ the temperature (310 K), and $\bar{p}_i$ the mean value of the probability of bin $i$ when averaged over the five blocks. The error in the $F_i$ estimate was calculated with $\sigma_{F_i} = kT \, \sigma_{\bar{p}_i}/\bar{p}_i$, where $\sigma_{\bar{p}_i}$ is twice the standard error of the mean of the probability. An offset was applied to the final PMF so that its lowest value was located at zero.

## qPCR quantification of recombinase activity

We grew 10 colonies of strain GY1761 carrying plasmid pGY466 overnight at 30°C in SD-L-W-M liquid cultures. The following day, we used these starter cultures to inoculate 12 ml of SD-W-M medium at OD$_{600}$ = 0.2. When monitoring growth by optical density measurements, we observed that it was fully exponential after 4 hr and until at least 8.5 hr. At 6.5 hr of growth, for each culture, we dispatched 0.1 ml in 96-well plate duplicates using one column (8 wells) per colony, stored aliquots by centrifuging 1 ml of the cell suspension at 3300 $g$ and freezing the cell pellet at $-20$°C ('exponential' negative control), and re-incubated the remaining of the culture at 30°C for later analysis at stationary phase. We exposed one plate (*Figure 4j* 'exponential' cyan samples) to blue light (PAUL apparatus, 460 nm, 3.63 mW/cm$^2$ intensity) for 40 min while the replicate plate was kept in the dark (*Figure 4j* 'exponential' gray samples). We pooled cells of the same column and stored them by centrifugation and freezing as above. The following day, we collected 1 ml of each saturated, frozen, and stored cells as above ('stationary' negative control). We dispatched the remaining of the cultures in a series of 96-well plates (0.1 ml/well, two columns per colony) and exposed these plates to blue light (PAUL apparatus, 460 nm, 3.63 mW/cm$^2$ intensity) for the indicated time (0, 2, 5, 10, 20, or 40 min). For each plate, following illumination, we collected and froze cells from six columns (*Figure 4h* samples) and re-incubated the plate in the dark for 90 min before collecting and freezing the remaining six columns (*Figure 4i*, x-axis samples). For genomic DNA extraction, we pooled cells from six wells of the same colony (one column), centrifuged and resuspended them in 280 µl in 50 mM EDTA, and added 20 µl of a 2 mg/ml Zymolyase stock solution (SEIKAGAKU, 20 U/mg) to the cell suspension and incubated it for 1 hr at 37°C for cell wall digestion. We then processed the digested cells with the Wizard Genomic DNA Purification Kit from Promega. We quantified DNA on a Nanodrop spectrophotometer and used ~100,000 copies of genomic DNA as template for qPCR, with primers 1P57 and 1P58 to amplify the edited target and primers 1B12 and 1C22 to amplify a control HMLalpha region that we used for normalization. We ran these reactions on a Rotorgene thermocycler (Qiagen). This allowed us to quantify the rate of excision of the floxed region as N$_{Lox}$/N$_{Total}$, where N$_{Lox}$ is the number of edited molecules and N$_{Total}$ the total number of DNA template molecules. To estimate N$_{Lox}$, we prepared mixtures of edited and non-edited genomic DNAs at known ratios of 0%, 0.5%, 1%, 5%, 10%, 50%, 70%, 90%, and 100% and applied (1P57, 1P58) qPCR using these mixtures as templates. This provided us with a standard curve that we then used to convert Ct values of the samples of interest into N$_{Lox}$ values. To estimate N$_{Total}$, we qPCR-amplified the HMLalpha control region from templates made of increasing concentrations of genomic DNA. We then used the corresponding standard curve to convert the Ct value of HMLalpha amplification obtained from the samples of interest into N$_{Total}$ values. No statistical power analysis was done prior to the experiment, no masking of samples IDs was applied during the experiments, and no data randomization was used.

## Acknowledgements

We thank Fabien Duveau for critical reading of the manuscript and for signal quantifications from microscopy images; Grégory Batt for fruitful discussions; Maria Teresa Texeira for strains; Sandrine Mouradian, Véronique Barateau, and SFR Biosciences Gerland-Lyon Sud (UMS344/US8) for access to flow cytometers and technical assistance; Christian Marchet for advice on LED controllers; and developers of R, Bioconductor, VMD, QLC+, and Ubuntu for their software. This work was supported by the European Research Council under the European Union's Seventh Framework Programme FP7/2007-2013 Grant Agreement no. 281359 and CNRS under the 'MITI 80 Prime' program grant READGEN.

## Additional information

### Competing interests

Hélène Duplus-Bottin, Martin Spichty, Gaël Yvert: A patent application covering LiCre and its potential applications has been filed. Ref: FR3079832 A1 and WO2019193205. Patent applicant: CNRS; inventors: Hélène Duplus-Bottin, Martin Spichty and Gaël Yvert. The other authors declare that no competing interests exist.

### Funding

| Funder | Grant reference number | Author |
| --- | --- | --- |
| H2020 European Research Council | StG-281359 (SiGHT) | Gaël Yvert |
| Centre National de la Recherche Scientifique | MITI 80 Prime READGEN | Gaël Yvert |

The funders had no role in study design, data collection and interpretation, or the decision to submit the work for publication.

### Author contributions

Hélène Duplus-Bottin, Investigation, Methodology, Constructed plasmids and strains, performed flow cytometry and yeast experiments; Martin Spichty, Software, Investigation, Visualization, Methodology, Performed PMF computations; Gérard Triqueneaux, Validation, Investigation, Methodology, Performed qPCR and human cell experiments, designed lentiviral vectors; Christophe Place, Resources, Methodology; Philippe Emmanuel Mangeot, Resources, Methodology, Designed and produced lentiviral vectors; Théophile Ohlmann, Franck Vittoz, Resources; Gaël Yvert, Conceptualization, Data curation, Software, Formal analysis, Supervision, Funding acquisition, Investigation, Visualization, Methodology, Writing - original draft, Project administration, Writing - review and editing

### Author ORCIDs

Hélène Duplus-Bottin (iD) https://orcid.org/0000-0003-2029-5646
Gaël Yvert (iD) https://orcid.org/0000-0003-1955-4786

### Decision letter and Author response

Decision letter https://doi.org/10.7554/eLife.61268.sa1
Author response https://doi.org/10.7554/eLife.61268.sa2

## Additional files

### Supplementary files

• Source code 1. R/Sweave code for analysis of flow-cytometry data.

• Supplementary file 1. File containing supplementary tables and supplementary text. Table S1: List of plasmids used in this study. Table S2: List of strains used in this study. Table S3: List of DNA

oligonucleotides used in this study. Supplementary Text S1: Synthetic nucleotidic sequences. Supplementary Text S2: Peptide sequences.

- Transparent reporting form

## Data availability

Raw flow-cytometry data have been deposited in Biostudies under accession code S-BSST580. Processed data used for figures are included in the supporting files.

The following dataset was generated:

| Author(s) | Year | Dataset title | Dataset URL | Database and Identifier |
|-----------|------|---------------|-------------|-------------------------|
| Yvert GI | 2021 | Light-inducible Cre recombinase (LiCre) | https://www.ebi.ac.uk/biostudies/studies/S-BSST580 | EBI Biostudies, S-BSST580 |

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
