## [Decision Letter]

**Acceptance summary:**

Your revisions nicely addressed the previously-stated concerns of the reviewers, particularly by supplying some much-needed direct comparisons across different systems with the new Figure 3. Additional detail on unsuccessful strategies and possible mechanisms were also improvements that strengthened this work as a manuscript which will be valued both for the generation of the specific LiCre reagent as well as the general findings of LOV-fusion protein design.

**Decision letter after peer review:**

Thank you for submitting your article "A monogenic and fast-responding Light-Inducible Cre recombinase as a novel optogenetic switch" for consideration by *eLife*. Your article has been reviewed by three peer reviewers, one of whom is a member of our Board of Reviewing Editors, and the evaluation has been overseen by Didier Stainier as the Senior Editor. The reviewers have opted to remain anonymous.

The reviewers have discussed the reviews with one another and the Reviewing Editor has drafted this decision to help you prepare a revised submission.

Summary:

This manuscript by Duplus-Bottin and co-workers describes the generation and characterization of LiCre, a blue light-inducible Cre recombinase. Using a combination of rational protein engineering and iterative screening, the authors generate fusions of the asLOV2 photosensory domain and Cre, settling a construct which fuses the LOV domain to the Cre N-terminus which has minimal dark state activity and substantial upregulation with illumination. LiCre is tested against several alternatives published by other groups, then demonstrated in applications are demonstrated in yeast and mammalian cell lines.

Overall, this manuscript provides a useful resource for several research communities via the generation of this tool, which exhibits several apparent advantages compared to the existing chemically- and light-controlled tools. Particular benefits are asserted for this single-chain light-regulated Cre, which are sensible. This said, the reviewers had substantial concerns regarding aspects of the comparisons of LiCre to existing tools, of mechanistic tests of the proposed LiCre regulatory models, and of details of the engineering of the successful LiCre construct (and some less effective variants). Those concerns will require a combination of new experimental data and manuscript edits to address, as detailed below, but the reviewers believe that the potential impact of this tool warrants the authors being given the opportunity to do so.

Essential revisions:

1) An important series of results are the comparisons to published tools in Figure 3C and D. While these data are compelling and were conducted under published conditions, it seems unfair to compare a fully optimized LiCre against the other systems which have had little to no optimization with the authors' experimental systems (and additionally, are illuminated with lower power levels). We encourage the authors to undertake some further optimization of the illumination protocols for the nMag/pMag and CRY2/Cib systems, as would be routine in most laboratories obtaining these tools for the first time, to provide a fairer comparison of their performance. It is essential to report all aspects of this optimization, particularly illumination power and duration/duty cycle, for readers to understand what has been done. These experiments should also provide controls for phototoxicity.

2) As a "Tools and Resources" article detailing the generation of a novel engineered protein, it is critical to fully describe the approach used for artificially regulating Cre. While some aspects of this are provided (Supplementary file 1, described in the Results), these could be clarified with a detailed schematic. Aside from translating the information provided in the supplementary text, it is quite unclear as to where the all-critical fusion point from AsLOV2 was between proteins were made.

3) Experiments here are strongly requested to test the mechanism of LiCre, which is discussed in the text but lacks in vitro experimental evidence to prop up the proposed model. In specific, the model predicts that fusion with AsLOV2 (and a corresponding competition between forming the AsLOV2 J(α) helix and the Cre A(α) helix) renders LiCre monomeric in darkness and tetrameric under blue light. Is there any experimental evidence supporting this (plausible) notion? Even if we take this model for granted, it is not clear whether the experiment with the two fluorescent reporters, described in the Results, is genuinely suited to discriminate between the two scenarios. A principal question is whether the two proposed scenarios are the only ones conceivable, or whether there could be alternative explanations. To distinguish between monomer activation and subsequent assembly, the authors should rather consider experiments at limiting light intensity. Monomer activation should scale linearly with light intensity but subsequent assembly of the synapse should scale with the fourth power. It should also be noted that this proposed artificial regulatory mode differs from conventional wisdom on the natural regulation of wildtype Cre – this protein doesn't form tetramers from monomers, it binds tightly to loxP sites and the loxP sites then associate to form a tetrameric assembly.

4) Several reviewers were confused about the discussion of a number of non-ideal or failed designs (Results), but it is unclear what benefits these bring to the manuscript. These could potentially provide useful information on the LOV-Cre integration, but more information will be needed than currently provided.

5) Finally, the manuscript would benefit from a thorough editing to make it more concise in parts. The Discussion section was highlighted by several reviewers as being unnecessarily lengthy, particularly in the discussions of the potential biotech applicability of these tools and in the light-activated Cre area more broadly. Aspects were also noted as overly repetitive from the Introduction.

---

## [Author Response]

Essential revisions:1) An important series of results are the comparisons to published tools in Figure 3C and D. While these data are compelling and were conducted under published conditions, it seems unfair to compare a fully optimized LiCre against the other systems which have had little to no optimization with the authors' experimental systems (and additionally, are illuminated with lower power levels). We encourage the authors to undertake some further optimization of the illumination protocols for the nMag/pMag and CRY2/Cib systems, as would be routine in most laboratories obtaining these tools for the first time, to provide a fairer comparison of their performance. It is essential to report all aspects of this optimization, particularly illumination power and duration/duty cycle, for readers to understand what has been done.

We have conducted these experiments and they are now presented in the revised manuscript. For this, we purchased and calibrated a custom computer-controlled LED spot which also enabled temperature-control of the illuminated cultures. We tested the efficiencies of LiCre, nMag/pMag and CRY2/Cib systems in parallel at various light intensities (Figure 3E). This showed that i) LiCre outperforms the other two systems at all intensity values, ii) the optimal intensity for nMag/pMag is as reported by the original authors and iii) the CRY2/Cib system does not work in these conditions either. We then applied periodic stimulations with different duty cycles on both LiCre and nMag/pMag split Cre at an intensity that was optimal for nMag/pMag (Figure 3F). This showed that the photocycle of LiCre is faster than the one of nMag/pMag, as expected from the different photocycle decay rates of AsLOV2 and VVD. We further characterized the response dynamics of LiCre at its optimal intensity and found a periodic regime conferring full activation with only 10% of illumination time.

These experiments should also provide controls for phototoxicity.

We quantified yeast viability by plating dilutions of illuminated cultures (full intensity under the LED spot) and scoring colony-forming units (Figure 3—figure supplement 1). There was no difference in CFUs between illuminated and control conditions. This result is mentioned in the revised main text.

2) As a "Tools and Resources" article detailing the generation of a novel engineered protein, it is critical to fully describe the approach used for artificially regulating Cre. While some aspects of this are provided (Supplementary file 1, described in the Results), these could be clarified with a detailed schematic. Aside from translating the information provided in the supplementary text, it is quite unclear as to where the all-critical fusion point from AsLOV2 was between proteins were made.

The full strategy to develop LiCre is now detailed in the revised manuscript. In particular:

– Results of all protein fusion designs are shown in Figure 2—figure supplements 1, 2, and 3.

– We have added negative results to Figure 2 (cases where photoactivation was not validated).

– The revised text (Results section) better describes the full approach.

– Peptidic sequences are explicitly provided in Supplementary file 1.

3) Experiments here are strongly requested to test the mechanism of LiCre, which is discussed in the text but lacks in vitro experimental evidence to prop up the proposed model. In specific, the model predicts that fusion with AsLOV2 (and a corresponding competition between forming the AsLOV2 J(α) helix and the Cre A(α) helix) renders LiCre monomeric in darkness and tetrameric under blue light. Is there any experimental evidence supporting this (plausible) notion? Even if we take this model for granted, it is not clear whether the experiment with the two fluorescent reporters, described in the Results, is genuinely suited to discriminate between the two scenarios. A principal question is whether the two proposed scenarios are the only ones conceivable, or whether there could be alternative explanations. To distinguish between monomer activation and subsequent assembly, the authors should rather consider experiments at limiting light intensity. Monomer activation should scale linearly with light intensity but subsequent assembly of the synapse should scale with the fourth power. It should also be noted that this proposed artificial regulatory mode differs from conventional wisdom on the natural regulation of wildtype Cre – this protein doesn't form tetramers from monomers, it binds tightly to loxP sites and the loxP sites then associate to form a tetrameric assembly.

We agree that the experiment with the two fluorescent proteins is not sufficient to address the mechanism of LiCre. We showed this experiment in the initial submission because it proved that the probability of recombination at one locus is neither independent nor fully associated with the probability of recombination at another locus. Then, when addressing point 1 above (Figure 3), we realized that illumination using a LED spot located above the cultures generated weaker LiCre induction than illumination in a LED box equipped with mirrors. It is therefore likely that illumination is not homogeneous in our assays, some cells receiving more light than others (we did not shake our cultures during illumination). This heterogeneity (extrinsic noise) could completely explain the two-colors experiment regardless of LiCre molecular mechanism: in any given cell, recombination at the two loci was stimulated by the same amount of light; this fact makes the two recombination events partially dependent. The experiment is therefore not necessarily informative and we removed it from the manuscript. Regarding the molecular mechanism of LiCre, we have revised the possible model more clearly and we moved it to the Discussion section and Figure 7. We distinguish Model 1, where activation of LiCre monomers allows them to bind to LoxP and then to assemble the tetramer, from Model 2, where LiCre is already bound to LoxP (as for Cre) in its dark state and photoactivation then enables tetramer assembly. Distinction between the two models depends on the affinity of dark-state LiCre for DNA. We have initiated experiments to address this but we are not ready to report any result yet. The scaling of LiCre response with light intensity is now shown in Figure 3E. The response seems to somewhat scale linearly with the x-axis, which is a logarithmic scale of light intensity. However, please note that data dispersion is substantial and other scalings could also match these measurements. Intermediate readouts of DNA binding and tetramer assembly are needed to clarify this mechanism in vitro and it will take time to set up and exploit such assays.

4) Several reviewers were confused about the discussion of a number of non-ideal or failed designs (Results), but it is unclear what benefits these bring to the manuscript. These could potentially provide useful information on the LOV-Cre integration, but more information will be needed than currently provided.

The revised manuscript now better describes the complete strategy that led to LiCre. Results of all fusions to VVD (first strategy) are displayed in Figure 2—figure supplement 1. Results of all fusions to AsLOV2 (second strategy), including failed designs, are displayed in Figure 2—figure supplements 2 and 3. Negative results that rejected candidate constructs are also shown: fusion at position 17 with wild-type Cre (Figure 2A) or at position 21 with Cre^E340A,D341A^ (Figure 2C), and the change of QIARDR to QILRDR in the LOV_Cre32 fusion (Figure 2B). The main text (Results section) was revised accordingly.

5) Finally, the manuscript would benefit from a thorough editing to make it more concise in parts. The Discussion section was highlighted by several reviewers as being unnecessarily lengthy, particularly in the discussions of the potential biotech applicability of these tools and in the light-activated Cre area more broadly. Aspects were also noted as overly repetitive from the Introduction.

The revised text is much more concise. We have deleted the entire "bioproduction" subsection of the Discussion section and we removed any redundancy with the Introduction.